# DiffusionNAG: Predictor-guided Neural Architecture Generation with Diffusion Models

**Sohyun An**[1]\*, **Hayeon Lee**[1]\*, **Jaehyeong Jo**[1], **Seanie Lee**[1], **Sung Ju Hwang**[1,2]
KAIST[1], DeepAuto.ai[2], Seoul, South Korea
{sohyunan, hayeon926, harryjo97, lsnfamily02, sjhwang82}@kaist.ac.kr

## Abstract

Existing NAS methods suffer from either an excessive amount of time for repetitive sampling and training of many task-irrelevant architectures. To tackle such limitations of existing NAS methods, we propose a paradigm shift from NAS to a novel conditional Neural Architecture Generation (NAG) framework based on diffusion models, dubbed DiffusionNAG. Specifically, we consider the neural architectures as directed graphs and propose a graph diffusion model for generating them. Moreover, with the guidance of parameterized predictors, DiffusionNAG can flexibly generate task-optimal architectures with the desired properties for diverse tasks, by sampling from a region that is more likely to satisfy the properties. This conditional NAG scheme is significantly more efficient than previous NAS schemes which sample the architectures and filter them using the property predictors. We validate the effectiveness of DiffusionNAG through extensive experiments in two predictor-based NAS scenarios: Transferable NAS and Bayesian Optimization (BO)-based NAS. DiffusionNAG achieves superior performance with speedups of up to $35\times$ when compared to the baselines on Transferable NAS benchmarks. Furthermore, when integrated into a BO-based algorithm, DiffusionNAG outperforms existing BO-based NAS approaches, particularly in the large MobileNetV3 search space on the ImageNet 1K dataset. Code is available at https://github.com/CownowAn/DiffusionNAG.

## 1 Introduction

While Neural Architecture Search (NAS) approaches automate neural architecture design, eliminating the need for manual design process with trial-and-error (Zoph & Le, 2017; Liu et al., 2019; Cai et al., 2019; Luo et al., 2018; Real et al., 2019; White et al., 2020), they mostly suffer from the high search cost, which often includes the full training with the searched architectures. To address this issue, many previous works have proposed to utilize parameterized property predictors (Luo et al., 2018; White et al., 2021a;b; 2023; Ning et al., 2020; 2021; Dudziak et al., 2020; Lee et al., 2021a;b; 2023a) that can predict the performance of an architecture without training. However, existing NAS approaches still result in large waste of time as they need to explore an extensive search space and the property predictors mostly play a passive role such as the evaluators that rank architecture candidates provided by a search strategy to simply filter them out during the search process.

To overcome such limitations, we propose a paradigm shift from NAS (Neural Architecture Search) to a novel conditional NAG (Neural Architecture Generation) framework that enables the *generation* of desired neural architectures. Specifically, we introduce a novel predictor-guided **Diffusion**-based **N**eural **A**rchitecture **G**enerative framework called DiffusionNAG, which explicitly incorporates the predictors into generating architectures that satisfy the objectives (e.g., high accuracy or robustness against attack). To achieve this goal, we employ the diffusion generative models (Ho et al., 2020; Song et al., 2021), which generate data by gradually injecting noise into the data and learning to reverse this process. They have demonstrated remarkable generative performance across a wide range of domains. Especially, we are inspired by their parameterized model-guidance mechanism (Sohl-Dickstein et al., 2015; Vignac et al., 2022) that allows the diffusion generative models to excel in conditional generation over diverse domains such as generating images that match specific labels (Ramesh et al., 2021) or discovering new drugs meeting particular property criteria (Lee et al., 2023b).

---

\*These authors contributed equally to this work.

In this framework, we begin by training the base diffusion generative model to generate architectures that follow the distribution of a search space without requiring expensive label information, e.g., accuracy. Then, to achieve our primary goal of generating architectures that meet the specified target condition, we deploy the trained diffusion model to diverse downstream tasks, while controlling the generation process with property predictors. Specifically, we leverage the gradients of parameterized predictors to guide the generative model toward the space of the architectures with desired properties. The proposed conditional NAG framework offers the key advantages compared with the conventional NAS methods as follows: Firstly, our approach facilitates efficient search by generating architectures that follow the specific distribution of interest within the search space, minimizing the time wasted exploring architectures that are less likely to have the desired properties. Secondly, DiffusionNAG, which utilizes the predictor for both NAG and evaluation purposes, shows superior performance compared to the traditional approach, where the same predictor is solely limited to the evaluation role. Lastly, DiffusionNAG is easily applicable to various types of NAS tasks (e.g., latency or robustness-constrained NAS) as we can swap out the predictors in a plug-and-play manner without retraining the base generative model, making it practical for diverse NAS scenarios.

Additionally, to ensure the generation of valid architectures, we design a novel score network for neural architectures. In previous works on NAS, neural architectures have been typically represented as directed acyclic graphs (Zhang et al., 2019) to model their computational flow where the input data sequentially passes through the multiple layers of the network to produce an output. However, existing graph diffusion models (Niu et al., 2020a; Jo et al., 2022) have primarily focused on *undirected* graphs, which represent structure information of graphs while completely ignoring the directional relationships between nodes, and thus cannot capture the computational flow in architectures. To address this issue, we introduce a score network that encodes the positional information of nodes to capture their order connected by directed edges.

We demonstrate the effectiveness of DiffusionNAG with extensive experiments under two key predictor-based NAS scenarios: 1) Transferable NAS and 2) Bayesian Optimization (BO)-based NAS. For Transferable NAS using transferable dataset-aware predictors, DiffusionNAG achieves superior or comparable performance with the speedup of up to 35× on four datasets from Transferable NAS benchmarks, including the large MobileNetV3 (MBv3) search space and NAS-Bench-201. Notably, DiffusionNAG demonstrates superior generation quality compared to MetaD2A (Lee et al., 2021a), a closely related *unconditional* generation-based method. For BO-based NAS with task-specific predictors, DiffusionNAG outperforms existing BO-based NAS approaches that rely on heuristic acquisition optimization strategies, such as random architecture sampling or architecture mutation, across four acquisition functions. This is because DiffusionNAG overcomes the limitation of existing BO-based NAS, which samples low-quality architectures during the initial phase, by sampling from the space of the architectures that satisfy the given properties. DiffusionNAG obtains especially large performance gains on the large MBv3 search space on the ImageNet 1K dataset, demonstrating its effectiveness in restricting the solution space when the search space is large. Furthermore, we verify that our score network generates 100% valid architectures by successfully capturing their computational flow, whereas the diffusion model for undirected graphs (Jo et al., 2022) almost fails.

Our contributions can be summarized as follows:

- We propose a paradigm shift from conventional NAS approaches to a novel conditional Neural Architecture Generation (NAG) scheme, by proposing a framework called DiffusionNAG. With the guidance of the property predictors, DiffusionNAG can generate task-optimal architectures for diverse tasks.

- DiffusionNAG offers several advantages compared with conventional NAS methods, including efficient and effective search, superior utilization of predictors for both NAG and evaluation purposes, and easy adaptability across diverse tasks.

- Furthermore, to ensure the generation of valid architectures by accurately capturing the computational flow, we introduce a novel score network for neural architectures that encodes positional information in directed acyclic graphs representing architectures.

- We have demonstrated the effectiveness of DiffusionNAG in Transferable NAS and BO-NAS scenarios, achieving significant acceleration and improved search performance in extensive experimental settings. DiffusionNAG significantly outperforms existing NAS methods in such experiments.

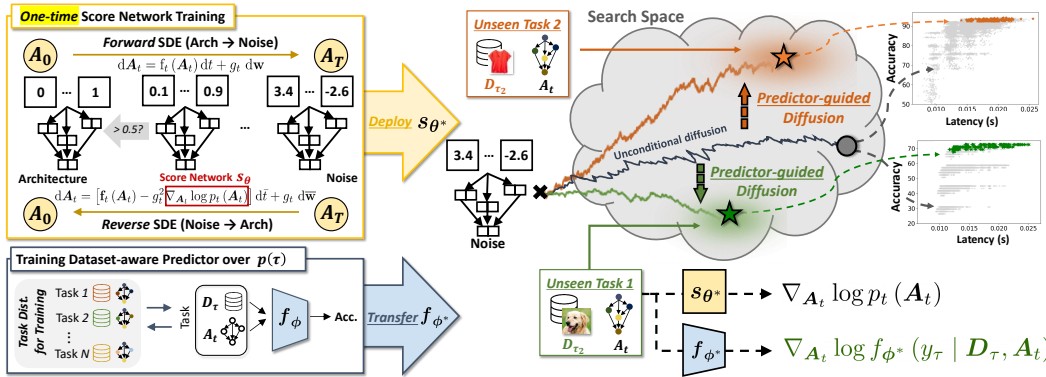

Figure 1: **Illustration of DiffusionNAG in Transferable NAS Scenarios.** DiffusionNAG generates desired neural architectures for a given unseen task by guiding the generation process with the transferable dataset-aware performance predictor $f_{\phi^*}(y_\tau | \boldsymbol{D}_\tau, \boldsymbol{A}_t)$.

## 2 METHOD

In Section 2.1, we first formulate the diffusion process for the generation of the architectures that follow the distribution of the search space. In Section 2.2, we propose a conditional diffusion framework for NAG that leverages a predictor for guiding the generation process. Finally, we extend the architecture generation framework for Transferable NAS in Section 2.3.

**Representation of Neural Architectures** A neural architecture $\boldsymbol{A}$ in the search space $\mathcal{A}$ is typically considered as a directed acyclic graph (DAG) (Dong & Yang, 2020b). Specifically, the architecture $\boldsymbol{A}$ with $N$ nodes is defined by its operator type matrix $\boldsymbol{\mathcal{V}} \in \mathbb{R}^{N \times F}$ and upper triangular adjacency matrix $\boldsymbol{\mathcal{E}} \in \mathbb{R}^{N \times N}$, as $\boldsymbol{A} = (\boldsymbol{\mathcal{V}}, \boldsymbol{\mathcal{E}}) \in \mathbb{R}^{N \times F} \times \mathbb{R}^{N \times N}$, where $F$ is the number of predefined operator sets. In the MobileNetV3 search space (Cai et al., 2020), $N$ represents the maximum possible number of layers, and the operation sets denote a set of combinations of the kernel size and width.

### 2.1 NEURAL ARCHITECTURE DIFFUSION PROCESS

As a first step, we formulate an unconditional neural architecture diffusion process. Following Song et al. (2021), we define a forward diffusion process that describes the perturbation of neural architecture distribution (search space) to a known prior distribution (e.g., Gaussian normal distribution) modeled by a stochastic differential equation (SDE), and then learn to reverse the perturbation process to sample the architectures from the search space starting from noise.

**Forward process** We define the forward diffusion process that maps the neural architecture distribution $p(\boldsymbol{A}_0)$ to the known prior distribution $p(\boldsymbol{A}_T)$ as the following Itô SDE:

$$\mathrm{d}\boldsymbol{A}_t = \mathbf{f}_t(\boldsymbol{A}_t)\mathrm{d}t + g_t\mathrm{d}\mathbf{w}, \tag{1}$$

where $t$-subscript represents a function of time ($F_t(\cdot) := F(\cdot, t)$), $\mathbf{f}_t(\cdot) \colon \mathcal{A} \to \mathcal{A}$ is the linear drift coefficient, $g_t \colon \mathcal{A} \to \mathbb{R}$ is the scalar diffusion coefficient, and $\mathbf{w}$ is the standard Wiener process. Following Jo et al. (2022), we adopt a similar approach where architectures are regarded as entities embedded in a continuous space. Subsequently, during the forward diffusion process, the architecture is perturbed with Gaussian noise at each step.

**Reverse process** The reverse-time diffusion process corresponding to the forward process is modeled by the following SDE (Anderson, 1982; Song et al., 2021):

$$\mathrm{d}\boldsymbol{A}_t = \left[\mathbf{f}_t(\boldsymbol{A}_t) - g_t^2 \nabla_{\boldsymbol{A}_t} \log p_t(\boldsymbol{A}_t)\right]\mathrm{d}\bar{t} + g_t\mathrm{d}\bar{\mathbf{w}}, \tag{2}$$

where $p_t$ denotes the marginal distribution under the forward diffusion process, $\mathrm{d}\bar{t}$ represents an infinitesimal negative time step and $\bar{\mathbf{w}}$ is the reverse-time standard Wiener process.

In order to use the reverse process as a generative model, the score network $\boldsymbol{s_\theta}$ is trained to approximate the score function $\nabla_{\boldsymbol{A}_t} \log p_t(\boldsymbol{A}_t)$ with the following score matching (Hyvärinen, 2005; Song

et al., 2021) objective, where $\lambda(t)$ is a given positive weighting function:

$$\boldsymbol{\theta}^* = \arg\min_{\boldsymbol{\theta}} \mathbb{E}_t \big\{ \lambda(t) \mathbb{E}_{\boldsymbol{A}_0} \mathbb{E}_{\boldsymbol{A}_t|\boldsymbol{A}_0} \| \boldsymbol{s}_{\boldsymbol{\theta}}(\boldsymbol{A}_t, t) - \nabla_{\boldsymbol{A}_t} \log p_t(\boldsymbol{A}_t) \|_2^2 \big\}. \tag{3}$$

Once the score network has been trained, we can generate neural architectures that follow the original distribution $p(\boldsymbol{A}_0)$ using the reverse process of Equation (2). To be specific, we start from noise sampled from the known prior distribution and simulate the reverse process, where the score function is approximated by the score network $\boldsymbol{s}_{\boldsymbol{\theta}^*}(\boldsymbol{A}_t, t)$. Following various continuous graph diffusion models (Niu et al., 2020a; Jo et al., 2022), we discretize the entries of the architecture matrices using the operator $\mathbb{1}_{>0.5}$ to obtain discrete 0-1 matrices after generating samples by simulating the reverse diffusion process. Empirically, we observed that the entries of the generated samples after simulating the diffusion process do not significantly deviate from integer values of 0 and 1.

**Score Network for Neural Architectures** To generate valid neural architectures, the score network should capture 1) the dependency between nodes, reflecting the computational flow (Dong & Yang, 2020a; Zhang et al., 2019), and 2) the accurate position of each layer within the overall architecture to comply with the rules of a specific search space. Inspired by Yan et al. (2021a) on architecture encoding, we use $L$ transformer blocks (T) with an attention mask $\boldsymbol{M} \in \mathbb{R}^{N \times N}$ that indicates the dependency between nodes, i.e., an upper triangular matrix of DAG representation (Dong & Yang, 2020a; Zhang et al., 2019), to parameterize the score network. (See Appendix B for more detailed descriptions) Furthermore, we introduce positional embedding $\mathbf{Emb}_{pos}(\mathbf{v}_i)$ to more accurately capture the topological ordering of layers in architectures, which leads to the generation of valid architectures adhering to specific rules within the given search space as follows:

$$\mathbf{Emb}_i = \mathbf{Emb}_{ops}(\mathbf{v}_i) + \mathbf{Emb}_{pos}(\mathbf{v}_i) + \mathbf{Emb}_{time}(t), \text{ where } \mathbf{v}_i : i\text{-th row of } \boldsymbol{\mathcal{V}} \text{ for } i \in [N], \tag{4}$$

$$\boldsymbol{s}_{\boldsymbol{\theta}}(\boldsymbol{A}_t, t) = \mathrm{MLP}(\boldsymbol{H}_L), \text{ where } \boldsymbol{H}_i^0 = \mathbf{Emb}_i, \boldsymbol{H}^l = \mathrm{T}(\boldsymbol{H}^{l-1}, \boldsymbol{M}) \text{ and } \boldsymbol{H}^l = [\boldsymbol{H}_1^l \cdots \boldsymbol{H}_N^l], \tag{5}$$

where $\mathbf{Emb}_{ops}(\mathbf{v}_i)$ and $\mathbf{Emb}_{time}(t)$ are embeddings of each node $\mathbf{v}_i$ and time $t$, respectively.

While simulating Equation (2) backward in time can generate random architectures within the entire search space, random generation is insufficient for the main goal of DiffusionNAG. Therefore, we introduce a *conditional* NAG framework to achieve this goal in the following section.

## 2.2 CONDITIONAL NEURAL ARCHITECTURE GENERATION

Inspired by the parameterized model-guidance scheme (Sohl-Dickstein et al., 2015; Vignac et al., 2022; Dhariwal & Nichol, 2021), we incorporate a parameterized predictor in our framework to actively guide the generation toward architectures that satisfy specific objectives. Let $y$ be the desired property (e.g., high accuracy or robustness against attacks) we want the neural architectures to satisfy. Then, we include the information of $y$ into the score function. To be specific, we generate neural architectures from the conditional distribution $p_t(\boldsymbol{A}_t|y)$ by solving the following conditional reverse-time SDE (Song et al., 2021):

$$\mathrm{d}\boldsymbol{A}_t = \big[\mathbf{f}_t(\boldsymbol{A}_t) - g_t^2 \nabla_{\boldsymbol{A}_t} \log p_t(\boldsymbol{A}_t|y)\big] \mathrm{d}\bar{t} + g_t \mathrm{d}\bar{\mathbf{w}}. \tag{6}$$

Here, we can decompose the conditional score function $\nabla_{\boldsymbol{A}_t} \log p_t(\boldsymbol{A}_t|y)$ in Equation (6) as the sum of two gradients that is derived from the Bayes' theorem $p(\boldsymbol{A}_t|y) \propto p(\boldsymbol{A}_t) p(y|\boldsymbol{A}_t)$:

$$\nabla_{\boldsymbol{A}_t} \log p_t(\boldsymbol{A}_t|y) = \nabla_{\boldsymbol{A}_t} \log p_t(\boldsymbol{A}_t) + \nabla_{\boldsymbol{A}_t} \log p_t(y|\boldsymbol{A}_t). \tag{7}$$

By approximating the score function $\nabla_{\boldsymbol{A}_t} \log p_t(\boldsymbol{A}_t)$ with the score network $\boldsymbol{s}_{\boldsymbol{\theta}^*}$, the conditional generative process of Equation (6) can be simulated if the term $\nabla_{\boldsymbol{A}_t} \log p_t(y|\boldsymbol{A}_t)$ could be estimated. Since $\log p_t(y|\boldsymbol{A}_t)$ represents the log-likelihood that the neural architecture $\boldsymbol{A}_t$ satisfies the target property $y$, we propose to model $\log p_t(y|\boldsymbol{A}_t)$ using a pre-trained predictor $f_{\boldsymbol{\phi}}(y|\boldsymbol{A}_t)$ parameterized by $\boldsymbol{\phi}$, which predicts the desired property $y$ given a perturbed neural architecture $\boldsymbol{A}_t$:

$$\nabla_{\boldsymbol{A}_t} \log p_t(y|\boldsymbol{A}_t) \approx \nabla_{\boldsymbol{A}_t} \log f_{\boldsymbol{\phi}}(y|\boldsymbol{A}_t). \tag{8}$$

As a result, we construct the guidance scheme with the predictor as follows, where $k_t$ is a constant that determines the scale of the guidance of the predictor:

$$\mathrm{d}\boldsymbol{A}_t = \big\{ \mathbf{f}_t(\boldsymbol{A}_t) - g_t^2 \big[ \boldsymbol{s}_{\boldsymbol{\theta}^*}(\boldsymbol{A}_t, t) + k_t \nabla_{\boldsymbol{A}_t} \log f_{\boldsymbol{\phi}}(y|\boldsymbol{A}_t) \big] \big\} \mathrm{d}\bar{t} + g_t \mathrm{d}\bar{\mathbf{w}}. \tag{9}$$

Intuitively, the predictor guides the generative process by modifying the unconditional score function which is estimated by $\boldsymbol{s}_{\boldsymbol{\theta}^*}$ at each sampling step. The key advantage of this framework is that we only need to train the score network **once** and can generate architectures with various target properties by simply changing the predictor. Our approach can reduce significant computational overhead for the conditional NAG compared to the classifier-free guidance scheme (Hoogeboom et al., 2022) that requires retraining the diffusion model every time the conditioning properties change.

Table 1: **Comparison with Transferable NAS on MBv3 Search Space.** The accuracies are reported with 95% confidence intervals over 3 runs. The *p-value* represents the result of a t-test conducted on the accuracies of 30 architecture samples obtained by our method and each baseline method.

| | Stats. | MetaD2A (Lee et al., 2021a) | Transferable NAS TNAS (Shala et al., 2023) | DiffusionNAG |
|---|---|---|---|---|
| CIFAR-10 | Max | 97.45±0.07 | 97.48±0.14 | **97.52±0.07** |
| | Mean | 97.28±0.01 | 97.22±0.00 | **97.39±0.01** |
| | Min | 97.09±0.13 | 95.62±0.09 | **97.23±0.06** |
| | *p-value* | 0.00000191 | 0.0024 | - |
| CIFAR-100 | Max | 86.00±0.19 | 85.95±0.29 | **86.07±0.16** |
| | Mean | 85.56±0.02 | 85.30±0.04 | **85.74±0.04** |
| | Min | 84.74±0.13 | 81.30±0.18 | **85.42±0.08** |
| | *p-value* | 0.0037 | 0.0037 | - |
| Aircraft | Max | 82.18±0.70 | **82.31±0.31** | 82.28±0.29 |
| | Mean | 81.19±0.11 | 80.86±0.15 | **81.47±0.05** |
| | Min | 79.71±0.54 | 74.99±0.65 | **80.88±0.54** |
| | *p-value* | 0.0169 | 0.0052 | - |
| Oxford-IIIT Pets | Max | 95.28±0.50 | 95.04±0.44 | **95.34±0.29** |
| | Mean | 94.55±0.03 | 94.47±0.10 | **94.75±0.10** |
| | Min | 93.68±0.16 | 92.39±0.04 | **94.28±0.17** |
| | *p-value* | 0.0025 | 0.0031 | - |

## 2.3 TRANSFERABLE CONDITIONAL NEURAL ARCHITECTURE GENERATION

Transferable NAS (Lee et al., 2021a; Shala et al., 2023) offers practical NAS capabilities for diverse real-world tasks, by simulating human learning. They acquire a knowledge from past NAS tasks to improve search performance on new tasks. In this section, to achieve highly efficient Transferable NAS, we extend the conditional NAG framework discussed earlier into a diffusion-based transferable NAG method by combining our framework with the transferable dataset-aware predictors from Transferable NAS methods (Lee et al., 2021a; Shala et al., 2023). A dataset-aware predictor $f_\phi(D, A_t)$ is conditioned on a dataset $D$. In other words, even for the same architecture, if datasets are different, the predictor can predict accuracy differently. The predictor is meta-learned with Equation (21) over the task distribution $p(\mathcal{T})$ utilizing a meta-dataset $\mathcal{S} := \{(A^{(i)}, y_i, D_i)\}_{i=1}^K$ with $K$ tasks consisting of (dataset, architecture, accuracy) triplets for each task. We use the meta-dataset collected by Lee et al. (2021a). The key advantage is that by exploiting the knowledge learned from the task distribution, we can conduct fast and accurate predictions for unseen datasets without additional predictor training. We integrate the meta-learned dataset-aware predictor $f_\phi(D, A_t)$ into the conditional neural architecture generative process (Equation (9)) for an unseen dataset $\tilde{D}$ as follows:

$$\mathrm{d}A_t = \left\{ \mathbf{f}_t(A_t) - g_t^2 \left[ s_{\theta^*}(A_t, t) + k_t \nabla_{A_t} \log f_\phi(y|\tilde{D}, A_t) \right] \right\} \mathrm{d}\bar{t} + g_t \mathrm{d}\bar{\mathbf{w}}. \quad (10)$$

## 3 EXPERIMENT

We validate the effectiveness of DiffusionNAG on two predictor-based NAS scenarios: Transferable NAS (Section 3.1) and BO-based NAS (Section 3.2). In Section 3.3, we demonstrate the effectiveness of the proposed score network.

**Search Space** We validate our framework on two Transferable NAS benchmark search spaces (Lee et al., 2021a): MobileNetV3 (MBv3) (Cai et al., 2020) and NAS-Bench-201 (NB201) (Dong & Yang, 2020b). Especially, MBv3 is a large search space, with approximately $10^{19}$ architectures. (Please see Appendix C.1 for detailed explanations.)

**Training Score Network** The score network is trained *only once* for all experiments conducted within each search space. Note that training the score network *only requires architectures (graph) without the need for accuracy which is expensive information*. The training process required 21.33 GPU hours (MBv3) and 3.43 GPU hours (NB201) on Tesla V100-SXM2, respectively.

## 3.1 COMPARISON WITH TRANSFERABLE NAS METHODS

**Experimental Setup** Transferable NAS methods (Shala et al., 2023; Lee et al., 2021a) are designed to leverage prior knowledge learned from previous NAS tasks, making NAS more practical on an

Table 2: **Comparison with Transferable NAS on NB201 Serach Space.** We present the accuracy achieved on four unseen datasets. Additionally, we provide the number of neural architectures (**Trained Archs**) that are actually trained to achieve accuracy. The accuracies are reported with 95% confidence intervals over 3 runs.

| Type | Method | CIFAR-10 Accuracy (%) | CIFAR-10 Trained Archs | CIFAR-100 Accuracy (%) | CIFAR-100 Trained Archs | Aircraft Accuracy (%) | Aircraft Trained Archs | Oxford-IIIT Pets Accuracy (%) | Oxford-IIIT Pets Trained Archs |
|---|---|---|---|---|---|---|---|---|---|
| | ResNet (He et al., 2016) | $93.97_{\pm0.00}$ | N/A | $70.86_{\pm0.00}$ | N/A | $47.01_{\pm1.16}$ | N/A | $25.58_{\pm3.43}$ | N/A |
| | RS (Bergstra & Bengio, 2012) | $93.70_{\pm0.36}$ | > 500 | $71.04_{\pm1.07}$ | > 500 | - | - | - | - |
| | REA (Real et al., 2019) | $93.92_{\pm0.30}$ | > 500 | $71.84_{\pm0.99}$ | > 500 | - | - | - | - |
| | REINFORCE (Williams, 1992) | $93.85_{\pm0.37}$ | > 500 | $71.71_{\pm1.09}$ | > 500 | - | - | - | - |
| One-shot NAS* | RSPS (Li & Talwalkar, 2019) | $84.07_{\pm3.61}$ | N/A | $52.31_{\pm5.77}$ | N/A | $42.19_{\pm3.88}$ | N/A | $22.91_{\pm1.65}$ | N/A |
| | SETN (Dong & Yang, 2019a) | $87.64_{\pm0.00}$ | N/A | $59.09_{\pm0.24}$ | N/A | $44.84_{\pm3.96}$ | N/A | $25.17_{\pm1.68}$ | N/A |
| | GDAS (Dong & Yang, 2019b) | $93.61_{\pm0.09}$ | N/A | $70.70_{\pm0.30}$ | N/A | $53.52_{\pm0.48}$ | N/A | $24.02_{\pm2.75}$ | N/A |
| | PC-DARTS (Xu et al., 2020) | $93.66_{\pm0.17}$ | N/A | $66.64_{\pm2.34}$ | N/A | $26.33_{\pm3.40}$ | N/A | $25.31_{\pm1.38}$ | N/A |
| | DrNAS (Chen et al., 2021) | $94.36_{\pm0.00}$ | N/A | $73.51_{\pm0.00}$ | N/A | $46.08_{\pm7.00}$ | N/A | $26.73_{\pm2.61}$ | N/A |
| BO-based NAS | BOHB (Falkner et al., 2018) | $93.61_{\pm0.52}$ | > 500 | $70.85_{\pm1.28}$ | > 500 | - | - | - | - |
| | GP-UCB | $\mathbf{94.37_{\pm0.00}}$ | 58 | $73.14_{\pm0.00}$ | 100 | $41.72_{\pm0.00}$ | 40 | $40.60_{\pm1.10}$ | 11 |
| | BANANAS (White et al., 2021a) | $\mathbf{94.37_{\pm0.00}}$ | 46 | $\mathbf{73.51_{\pm0.00}}$ | 88 | $41.72_{\pm0.00}$ | 40 | $40.15_{\pm1.59}$ | 17 |
| | NASBOWL (Ru et al., 2021) | $94.34_{\pm0.00}$ | 100 | $\mathbf{73.51_{\pm0.00}}$ | 87 | $53.73_{\pm0.83}$ | 40 | $41.29_{\pm1.10}$ | 17 |
| | HEBO (Cowen-Rivers et al., 2022) | $94.34_{\pm0.00}$ | 100 | $72.62_{\pm0.20}$ | 100 | $49.32_{\pm6.10}$ | 40 | $40.55_{\pm1.15}$ | 18 |
| Transferable NAS | TNAS (Shala et al., 2023) | $\mathbf{94.37_{\pm0.00}}$ | 29 | $\mathbf{73.51_{\pm0.00}}$ | 59 | $\mathbf{59.15_{\pm0.58}}$ | 26 | $40.00_{\pm0.00}$ | 6 |
| | MetaD2A (Lee et al., 2021a) | $\mathbf{94.37_{\pm0.00}}$ | 100 | $73.34_{\pm0.04}$ | 100 | $57.71_{\pm0.20}$ | 40 | $39.04_{\pm0.20}$ | 40 |
| | DiffusionNAG (Ours) | $\mathbf{94.37_{\pm0.00}}$ | 1 | $\mathbf{73.51_{\pm0.00}}$ | 2 | $58.83_{\pm3.75}$ | 3 | $\mathbf{41.80_{\pm3.82}}$ | 2 |

* We report the search time of one-shot NAS methods in Appendix C.3.

Table 3: **Statistics of the generated architectures.** Each method generates 1,000 architectures.

| Target Dataset | Stats. | Oracle Top-1,000 | Random | MetaD2A | Uncond. + Sorting | Cond. (Ours) |
|---|---|---|---|---|---|---|
| CIFAR10 | Max | 94.37 | **94.37** | **94.37** | 94.37 | 94.37 |
| | Mean | 93.50 | 87.12 | 91.52 | 90.77 | **93.13** |
| | Min | 93.18 | 10.00 | 10.00 | 10.00 | **86.44** |
| CIFAR100 | Max | 73.51 | 72.74 | **73.51** | 73.16 | 73.51 |
| | Mean | 70.62 | 61.59 | 67.14 | 66.37 | **70.34** |
| | Min | 69.91 | 1.00 | 1.00 | 1.00 | **58.09** |

Figure 2: **The distribution of generated architectures.**

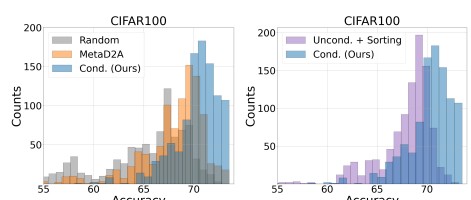

unseen task. To achieve this, all Transferable NAS methods, including our DiffusionNAG, utilize a *transferable dataset-aware accuracy predictor*, as described in Section 2.3. The dataset-aware predictor is meta-trained on the meta-dataset provided by Lee et al. (2021a), which consists of 153,408/4,230 meta-training tasks for MBv3/NB201, respectively. For more details, please refer to Lee et al. (2021a). **MetaD2A** (Lee et al., 2021a), which is the most closely related to our work, includes an unconditional architecture generative model that *explicitly excludes the dataset-aware predictor* during the generation process. Instead, MetaD2A needs to search for optimal architectures across multiple tasks, train these architectures to obtain their accuracy data, and use this costly accuracy collection to train its generative model. Besides, it uses the dataset-aware predictor only during the subsequent evaluation stage to rank the generated architectures. During the test phase, it first *objective-unconditionally generates* architectures and then evaluates the top architectures using its predictor. **TNAS** (Shala et al., 2023) enhances the meta-learned dataset-aware predictor's adaptability to unseen datasets by utilizing BO with the deep-kernel GP strategy without involving any generation process (Please see Appendix C.2 for details of the baselines.). **DiffusionNAG** *conditionally generates* architectures with the diffusion model guided by the dataset-aware predictor. Our generation process, with a sampling batch size of 256, takes up to 2.02 GPU minutes on Tesla V100-SXM2 to sample one batch. Finally, we select the top architectures sorted by the predictor among the generated candidates. We conduct experiments on Transferable NAS benchmarks (Lee et al., 2021a) such as four unseen datasets - CIFAR-10, CIFAR-100, Aircraft (Maji et al., 2013), and Oxford IIT Pets (Parkhi et al., 2012) from large search space MBv3 (Table 1) and, NB201 (Table 2).

**Results on MBv3 Search Space** In Table 1, MetaD2A, TNAS, and DiffusionNAG obtain the top 30 neural architectures for each datasets. Subsequently, we train these architectures on the datasets following the training pipeline described in Appendix C.5. Once the architectures are trained, we analyze the accuracy statistics for each method's group of architectures. Additionally, we calculate *p-value* to assess the statistical significance of performance differences between the architecture groups obtained via DiffusionNAG and each method. A *p-value* of 0.05 or lower denotes that a statistically meaningful difference exists in the performances of the generated architectures between the two groups.

The results demonstrate that, except for the Aircraft dataset, DiffusionNAG consistently provides architectures with superior maximum accuracy (**max**) compared to other methods across three

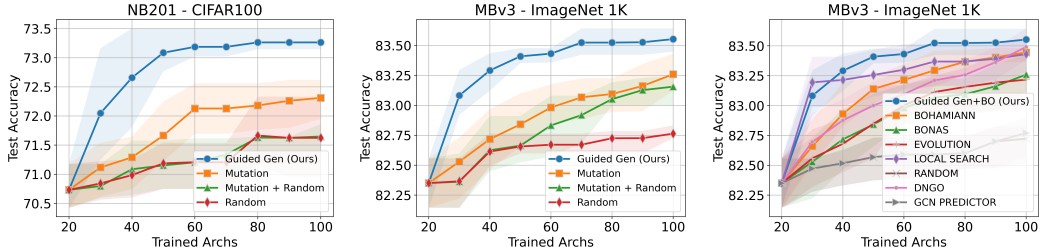

Figure 3: **Comparison Results on Existing AO Strategies.** *Guided Gen (Ours)* strategy provides a pool of candidate architectures, guiding them toward a high-performance distribution using the current population with DiffusionNAG. We report the results of multiple experiments with 10 different random seeds.

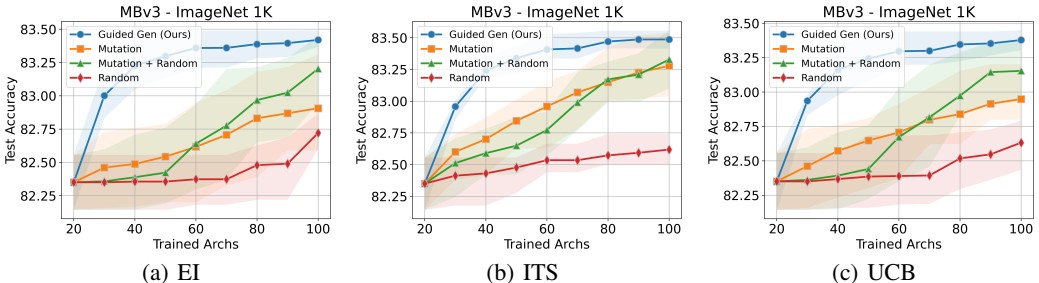

Figure 4: **Experimental Results on Various Acquisition Functions.** *Ours* consistently outperforms the heuristic approaches on various acquisition functions. We run experiments with 10 different random seeds.

datasets. Additionally, the **mean** accuracy and minimum accuracy (**min**) of architectures within the DiffusionNAG group are higher across all datasets. In particular, the *p-values* obtained from comparing the groups of architectures suggested by DiffusionNAG and those from other baselines are consistently below the 0.05 threshold across all datasets. This indicates that the architectures generated by DiffusionNAG have shown statistically significant performance improvements compared to those provided by the baseline methods when using transferable dataset-aware predictors. Furthermore, the results clearly support the superiority of the proposed predictor-guided conditional architecture generation method compared with either excluding predictors during generation (MetaD2A) or relying solely on predictors without generating architectures (TNAS).

**Results on NB201 Search Space**   We highlight two key aspects from the results of Table 2. Firstly, the architectures generated by DiffusionNAG attain oracle accuracies of **94.37**% and **73.51**% on CIFAR-10 and CIFAR-100 datasets, respectively, and outperform architectures obtained by the baseline methods on Aircraft and Oxford-IIIT Pets datasets. While MetaD2A and TNAS achieve accuracies of **59.15**%/**57.71**% and **40.00**%/**39.04**% on Aircraft and Oxford-IIIT Pets datasets, respectively, DiffusionNAG achieves comparable or better accuracies of **58.83**% and **41.80**%, demonstrating its superiority. Secondly, DiffusionNAG significantly improves the search efficiency by minimizing the number of architectures that require full training (**Trained Archs**) to obtain a final accuracy (For CIFAR-10 and CIFAR-100, an accuracy is retrieved from NB201 benchmarks) compared to all baselines. Specifically, when considering the Aircraft and Oxford-IIIT Pets datasets, DiffusionNAG only needs to train **3**/**2** architectures for each dataset to complete the search process while MetaD2A and TNAS require **40**/**40** and **26**/**6** architectures, respectively. This results in a **remarkable speedup** of at least **15**× and up to **35**× on average.

**Further Anaylsis**   We further analyze the accuracy statistics of the distribution of architectures generated by each method within the NB201 search space. Specifically, we conduct an in-depth study by generating 1,000 architectures using each method and analyzing their distribution, as presented in Table 3 and Figure 2. We compare DiffusionNAG with two other methods: random architecture sampling (**Random**) and **MetaD2A**. Additionally, to assess the advantage of using a predictor in both the NAG and evaluation phases compared to an approach where the predictor is solely used in the evaluation phase, we unconditionally generate 10,000 architectures and then employ the predictor to select the top 1,000 architectures (**Uncond. + Sorting**). DiffusionNAG (**Cond.**) leverages the dataset-aware predictor $f_\phi(D, A_t)$ to guide the generation process following Equation (10).

The results from Table 3 and Figure 2 highlight three key advantages of our model over the baselines. Firstly, our model generates a higher proportion of high-performing architectures for each target dataset, closely following the Oracle Top-1000 distribution within the search space. Secondly, our model avoids generating extremely low-accuracy architectures, unlike the baseline methods, which generate architectures with only 10% accuracy. This suggests that our model is capable of focusing on a target architecture distribution by excluding underperforming architectures. Lastly, as shown in Figure 2, DiffusionNAG (**Cond.**) outperforms sorting after the unconditional NAG process (**Uncond. + Sorting**). These results highlight the value of involving the predictor not only in the evaluation phase but also in the NAG process, emphasizing the necessity of our *conditional NAG* framework.

### 3.2 Improving Existing Bayesian Optimization-based NAS

In this section, we have demonstrated that DiffusionNAG significantly outperforms existing heuristic architecture sampling techniques used in Bayesian Optimization (BO)-based NAS approaches, leading to improved search performance in BO-based NAS.

**BO-based NAS** The typical BO algorithm for NAS (White et al., 2023) is as follows: 1) Start with an initial population containing neural architecture-accuracy pairs by uniformly sampling $n_0$ architectures and obtaining their accuracy. 2) Train a predictor using architecture-accuracy pairs in the population, and 3) Sample $c$ candidate architectures by the Acquisition Optimization strategy (**AO strategy**) (White et al., 2021a) and choose the one maximizing an acquisition function based on the predictions of the predictor. 4) Evaluate the accuracy of the selected architecture after training it and add the pair of the chosen architecture and its obtained accuracy to the population. 5) Repeat steps 2) to 4) during $N$ iterations, and finally, select the architecture with the highest accuracy from the population as the search result. (For more details, refer to Algorithm 1 in the Appendix.)

Our primary focus is on replacing the existing **AO strategy** in step 3) with DiffusionNAG to improve the search efficiency of BO-based NAS approaches. **Baseline AO Strategy**: The simplest AO strategy is randomly sampling architecture candidates (**Random**). Another representative AO strategy is **Mutation**, where we randomly modify one operation in the architecture with the highest accuracy in the population. **Mutation + Random** combines two aforementioned approaches. **Guided Gen (Ours)**: Instead of relying on these heuristic strategies, we utilize DiffusionNAG to generate the candidate architectures. Specifically, we train a predictor $f_\phi(y|\boldsymbol{A}_t)$, as described in Equation (8), using architecture-accuracy pairs in the population. The trained predictor guides the generation process of our diffusion model to generate architectures. We then provide these generated architecture candidates to the acquisition function in step 3) (See Algorithm 2 in the Appendix.)

**Comparison Results with Existing AO Strategies** The left and middle sides in Figure 3 illustrates our comparison results with existing AO strategies. These results clearly highlight the effectiveness of DiffusionNAG (**Guided Gen (Ours)**), as it significantly outperforms existing AO strategies such as **Random**, **Mutation**, and **Mutation + Random** on the CIFAR100 dataset from NB201 and the large-scale ImageNet 1K (Deng et al., 2009) dataset within the extensive MBv3 search space. In particular, BO-based NAS methods employing **Random** or **Mutation** strategies often suffer from the issue of wasting time on sampling low-quality architectures during the initial phase (White et al., 2020; Zela et al., 2022). In contrast, DiffusionNAG effectively addresses this issue by offering relatively high-performing architectures right from the start, resulting in a significant reduction in search times. As a result, as shown in the right side of Figure 3, our approach outperforms existing BO-based NAS methods, by effectively addressing the search cost challenge of them.

**Experimental Results on Various Acquisition Functions** In addition to the Probability of Improvement (**PI**) used in Figure 3, we investigate the benefits of DiffusionNAG across various acquisition functions, such as Expected Improvement (**EI**), Independent Thompson sampling (**ITS**), and Upper Confidence Bound (**UCB**) as shown in Figure 4. (Please see Appendix D.2 for more details on acquisition functions.). The experimental results verify that DiffusionNAG (**Ours**) consistently outperforms heuristic approaches, including **Mutation**, **Random**, and **Mutation + Random** approaches, on four acquisition functions: **PI**, **EI**, **ITS**, and **UCB**, in the large MBv3 search space.

### 3.3 The Effectiveness of Score Network for Neural Architectures

In this section, we validate the ability of the proposed score network to generate architectures that follow the distribution of NB201 and MBv3 search spaces. For NB201, we construct the training set

Table 4: **Generation Quality.** We generate 1,000 samples with each method for 3 runs of different seeds.

| Method | NAS-Bench-201 | | | MobileNetV3 | | |
|---|---|---|---|---|---|---|
| | Validity (%) ↑ | Uniq. (%) ↑ | Novelty (%) ↑ | Validity (%) ↑ | Uniq. (%) ↑ | Novelty (%) ↑ |
| GDSS Jo et al. (2022) | $4.56_{\pm 1.44}$ | - | - | $0.00_{\pm 0.00}$ | - | - |
| Ours (w/o Pos. Emb.) | $\mathbf{100.00}_{\pm 0.00}$ | $\mathbf{98.96}_{\pm 0.49}$ | $49.08_{\pm 2.05}$ | $42.17_{\pm 1.80}$ | $\mathbf{100.00}_{\pm 0.00}$ | $\mathbf{100.00}_{\pm 0.00}$ |
| Ours (w/ Pos. Emb.) | $\mathbf{100.00}_{\pm 0.00}$ | $98.70_{\pm 0.66}$ | $\mathbf{49.20}_{\pm 1.96}$ | $\mathbf{100.00}_{\pm 0.00}$ | $\mathbf{100.00}_{\pm 0.00}$ | $\mathbf{100.00}_{\pm 0.00}$ |

by randomly selecting 50% of the architectures from the search space, while for MBv3, we randomly sample 500,000 architectures. We evaluate generated architectures using three metrics (Zhang et al., 2019): **Validity**, **Uniqueness**, and **Novelty**. **Validity** measures the proportion of valid architectures generated by the model, **Uniqueness** quantifies the proportion of unique architectures among the valid ones, and **Novelty** indicates the proportion of valid architectures that are not present in the training set. As shown in Table 4, our score network generates valid architectures with **100% Validity**, whereas **GDSS** (Jo et al., 2022), a state-of-the-art graph diffusion model designed for *undirected* graphs, fails to generate valid architectures, with the validity of only **4.56%** and **0.00%** for NB201 and MBv3, respectively. Furthermore, our positional embedding yields significant improvements, indicating that it successfully captures the topological ordering of nodes within the architectures. Notably, in the MBv3, **Validity** improves from **42.17%** to **100.00%**, highlighting the necessity of positional embedding for generating architectures with a large number of nodes (a.k.a. "long-range"). Additionally, our framework generates **49.20%/100.00%** novel architectures that are not found in the training set, as well as unique architectures **98.70%/100.00%** for NB201 and MBv3, respectively.

## 4 RELATED WORK

**Neural Architecture Search** NAS is an automated architecture search process (Ning et al., 2021; Zoph & Le, 2017) and roughly can be categorized into reinforcement learning-based (Zoph & Le, 2017; Zoph et al., 2018; Pham et al., 2018), evolutionary algorithm-based (Real et al., 2019; Lu et al., 2020), and gradient-based methods (Luo et al., 2018; Liu et al., 2019; Dong & Yang, 2019b; Xu et al., 2020; Chen et al., 2021). Recently, Shala et al. (2023); Lee et al. (2021a) have proposed Transferable NAS to rapidly adapt to unseen tasks by leveraging prior knowledge. However, they still suffer from the high search cost. DiffusionNAG addresses these limitations by generating architectures satisfying the objective with a guidance scheme of a meta-learned dataset-aware predictor.

**Diffusion Models** Diffusion models, as demonstrated in prior work (Song & Ermon, 2019; Ho et al., 2020; Song et al., 2021), are designed to reverse the data perturbation process, enabling them to generate samples from noisy data. They have achieved success in a variety of domains, including images (Nichol et al., 2022; Rombach et al., 2022), audio (Jeong et al., 2021; Kong et al., 2021), and graphs (Niu et al., 2020b; Jo et al., 2022). However, existing diffusion models are not well-suited for Neural Architecture Generation (NAG) because their primary focus is on unconditionally generating undirected graphs. To overcome this limitation, this study introduces a conditional diffusion-based generative framework tailored for generating architectures represented as directed acyclic graphs that meet specified conditions, such as accuracy requirements.

## 5 CONCLUSION

This study introduced a novel conditional Neural Architecture Generation (NAG) framework called DiffusionNAG, which is the paradigm shift from existing NAS methods by leveraging diffusion models. With the guidance of a property predictor for a given task, DiffusionNAG can efficiently generate task-optimal architectures. Additionally, the introduction of a score network ensures the generation of valid neural architectures. Extensive experiments under two key predictor-based NAS scenarios demonstrated that DiffusionNAG outperforms existing NAS methods, especially effective in the large search space. We believe that our success underscores the potential for further advancements in NAS methodologies, promising accelerated progress in the development of optimal neural architectures.

**Reproducibility Statement**  To ensure reliable and reproducible results, we have provided the source code, available at https://github.com/CownowAn/DiffusionNAG, and the "NAS Best Practices Checklist" in Appendix A. Furthermore, we have described the details of the diffusion process and models used in DiffusionNAG in Appendix B, as well as the experimental details in Appendix C.

**Ethics Statement**  This paper introduces a novel conditional Neural Architecture Generation framework called DiffusionNAG, aimed at advancing the field of generative models, especially for neural architectures. We believe that our work can open up new avenues for neural architecture search/generation, achieving significant acceleration and improved search performance in extensive experimental settings. Although the possibility of causing any negative societal impacts through our work is low, we should be cautious about the potential for our work to be utilized to generate neural architectures for unethical purposes.

**Acknowledgement**  This work was supported by the National Research Foundation of Korea(NRF) grant funded by the Korea government(MSIT) (No. RS-2023-00256259), Center for Applied Research in Artificial Intelligence (CARAI) grant funded by DAPA and ADD (UD190031RD), Institute of Information & communications Technology Planning & Evaluation (IITP) grant funded by the Korea government(MSIT) (No.2022-0-00713), and Institute of Information & communications Technology Planning & Evaluation (IITP) grant funded by the Korea government(MSIT) (No.2019-0-00075, Artificial Intelligence Graduate School Program(KAIST)).

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

**Organization**   This supplementary file contains detailed explanations of the materials that are not covered in the main paper, along with additional experimental results. The supplementary file is organized as follows:

- **Appendix A** - We cover the points in the NAS Best Practices Checklist (Lindauer & Hutter, 2020).
- **Appendix B** - We elaborate on the details of our approach, DiffusionNAG.
- **Appendix C** - We elaborate on the detailed experiment setups, such as search space, baselines, datasets, architecture training pipeline, and implementation details.
- **Appendix D** - We provide the results of additional experiments.

## A   NAS BEST PRACTICE CHECKLIST

In this section, we provide a description of how we cover each point in the NAS Best Practice Checklist (Lindauer & Hutter, 2020).

1. Best Practices for Releasing Code For all experiments you report:
   (a) Did you release the code for the training pipeline used to evaluate the final architectures? We released the code for the training pipeline.
   (b) Did you release the code for the search space? In our main experiments, we utilized the NAS-Bench-201 and MobileNetV3 search space, which is publicly available along with its description and code.
   (c) Did you release the hyperparameters used for the final evaluation pipeline, as well as random seeds? Yes, it can be found in the code we provide.
   (d) Did you release code for your NAS method? The code for our work is available on GitHub: https://github.com/CownowAn/DiffusionNAG.
   (e) Did you release hyperparameters for your NAS method, as well as random seeds? Yes, it can be found in the code we provide.
2. Best practices for comparing NAS methods
   (a) For all NAS methods you compare, did you use exactly the same NAS benchmark, including the same dataset (with the same training-test split), search space and code for training the architectures and hyperparameters for that code? Yes, for a fair comparison, we used the same evaluation pipeline for all NAS methods we compare.
   (b) Did you control for confounding factors (different hardware, versions of DL libraries, different runtimes for the different methods)? To control for confounding factors, we ensured that all methods were executed on the same hardware and environment.
   (c) Did you run ablation studies? Yes.
   (d) Did you use the same evaluation protocol for the methods being compared? Yes.
   (e) Did you compare performance over time? Yes.
   (f) Did you compare to random search? Performance comparison to random search and other baselines can be found in Table 2 of the main paper.
   (g) Did you perform multiple runs of your experiments and report seeds? For each of the experiments we performed three runs with different seeds (777,888,999).
   (h) Did you use tabular or surrogate benchmarks for in-depth evaluations? We used NAS-Bench-201 (Dong & Yang, 2020b) as a tabular benchmark.
3. Best practices for reporting important details
   (a) Did you report how you tuned hyperparameters, and what time and resources this required? Yes.
   (b) Did you report the time for the entire end-to-end NAS method (rather than, e.g., only for the search phase)? Yes.
   (c) Did you report all the details of your experimental setup? The details of our experimental setup can be found in Appendix C.

## B   DETAILS OF DIFFUSIONNAG

### B.1   OVERVIEW

In this section, we first describe the forward diffusion process that perturbs neural architectures towards random graphs in Appendix B.2. Subsequently, we illustrate our specialized score network for estimating neural architecture scores in Appendix B.3. Finally, we elaborate the components of the dataset-aware predictor in DiffusionNAG in Appendix B.4.

## B.2 Diffusion Process

Our approach aims to apply continuous-time generative diffusion processes to neural architectures. Specifically, we utilize the Variance Exploding (VE) SDE (Song et al., 2021) to model the forward diffusion process for our neural architecture generation framework. The following SDE describes the process of the VE SDE (Jo et al., 2022):

$$\mathrm{d}\boldsymbol{A} = \sigma_{\min} \left( \frac{\sigma_{\max}}{\sigma_{\min}} \right)^t \sqrt{2 \log \frac{\sigma_{\max}}{\sigma_{\min}}} \mathrm{d}\mathbf{w}, \tag{11}$$

where $\sigma_{\min}$, $\sigma_{\max}$, and $t \in (0,1]$ are hyperparameters. The transition distribution of the process is Gaussian, given that Equation (11) has a linear drift coefficient. We can obtain the mean and covariance using the result of Equation (5.50) and (5.51) of Särkkä & Solin (2019) as follows:

$$p_{t|0}(\boldsymbol{A}_t|\boldsymbol{A}_0) = \mathcal{N}\left( \boldsymbol{A}_t;\ \boldsymbol{A}_0\ , \sigma_{\min}^2 \left( \frac{\sigma_{\max}}{\sigma_{\min}} \right)^{2t} \mathbf{I} \right). \tag{12}$$

During the forward diffusion process, Gaussian noise is applied to perturb the architecture, following the noise scheduling modeled by the prescribed SDE. In order to employ the reverse process, which corresponds to the forward process, as a generative model, the score network $s_{\boldsymbol{\theta}}$ is trained to approximate the score function $\nabla_{\boldsymbol{A}_t} \log p_t(\boldsymbol{A}_t)$.

## B.3 Score Network for Neural Architectures

As proposed in Section 2.1 of the main paper, we introduce a score network specifically designed for neural architectures. The distinctive aspect of our score network is the incorporation of positional embedding for nodes within directed acyclic graphs (DAGs), which allows for capturing the topological ordering of neural architectures. The input embedding and computation of this score network are as follows:

$$\mathbf{Emb}_i = \mathbf{Emb}_{ops}\left(\mathbf{v}_i\right) + \mathbf{Emb}_{pos}\left(\mathbf{v}_i\right) + \mathbf{Emb}_{time}\left(t\right), \text{ where } \mathbf{v}_i : i\text{-th row of } \boldsymbol{\mathcal{V}} \text{ for } i \in [N], \tag{13}$$

$$\boldsymbol{s}_{\boldsymbol{\theta}}\left(\boldsymbol{A}_t, t\right) = \mathrm{MLP}\left(\boldsymbol{H}_L\right), \text{ where } \boldsymbol{H}_i^0 = \mathbf{Emb}_i, \boldsymbol{H}^l = \mathrm{T}\left(\boldsymbol{H}^{l-1}, \boldsymbol{M}\right) \text{ and } \boldsymbol{H}^l = [\boldsymbol{H}_1^l \cdots \boldsymbol{H}_N^l]^\top. \tag{14}$$

Here, we denote $\mathbf{Emb}_{ops}\left(\mathbf{v}_i\right)$ as the embedding of each node (operation) $\mathbf{v}_i$, and $\mathbf{Emb}_{time}\left(\mathbf{v}_i\right)$ as the embedding of time $t$. Additionally, we introduce the positional embedding $\mathbf{Emb}_{pos}\left(\mathbf{v}_i\right)$ to incorporate positional information of the node into the input embeddings. Inspired by the previous work (Yan et al., 2021a), we employ $L$ transformer blocks (T) (Vaswani et al., 2017) to parameterize the score network. These blocks utilize an attention mask $\boldsymbol{M} \in \mathbb{R}^{N \times N}$, where $N$ represents the number of nodes in the architecture. In our approach, a pair of nodes (operations) in the architecture is considered dependent if a directed edge is connected. The attention mask $\boldsymbol{M}$ in the transformer block is determined by the value of the adjacency matrix $\boldsymbol{\mathcal{E}}$ of a neural architecture $\boldsymbol{A}$ following the equation:

$$\boldsymbol{M}_{i,j} = \begin{cases} 0, & \text{if} \quad \boldsymbol{\mathcal{E}}_{i,j} = 1 \\ -10^9, & \text{if} \quad \boldsymbol{\mathcal{E}}_{i,j} = 0 \end{cases} \tag{15}$$

Each Transformer block (T) consists of $n_{\mathrm{head}}$ attention heads. The computation of the $l$-th Transformer block is described as follows:

$$\boldsymbol{Q}_p = \boldsymbol{H}^{l-1}\boldsymbol{W}_{qp}^l, \boldsymbol{K}_p = \boldsymbol{H}^{l-1}\boldsymbol{W}_{kp}^l, \boldsymbol{V}_p = \boldsymbol{H}^{l-1}\boldsymbol{W}_{vp}^l \tag{16}$$

$$\hat{\boldsymbol{H}}_p^l = \mathrm{softmax}\left( \frac{\boldsymbol{Q}_p\boldsymbol{K}_p^\top}{\sqrt{d_h}} + \boldsymbol{M} \right) \boldsymbol{V}_p \tag{17}$$

$$\hat{\boldsymbol{H}}^l = \mathrm{concatenate}\left( \hat{\boldsymbol{H}}_1^l, \hat{\boldsymbol{H}}_2^l, \ldots, \hat{\boldsymbol{H}}_{n_{\mathrm{head}}}^l \right) \tag{18}$$

$$\boldsymbol{H}^l = \mathrm{ReLU}\left( \hat{\boldsymbol{H}}^l\boldsymbol{W}_1^l + \boldsymbol{b}_1^l \right) \boldsymbol{W}_2^l + \boldsymbol{b}_2^l \tag{19}$$

where the size of $\boldsymbol{H}^l$ and $\hat{\boldsymbol{H}}_p^l$ is $N \times d_h$ and $N \times (d_h/n_{\mathrm{head}})$, respectively. Additionally, in the attention operation of the $p$-th head, we denote $\boldsymbol{Q}_p$, $\boldsymbol{K}_p$, and $\boldsymbol{V}_p$ as the "Query", "Key", and "Value" matrices, respectively. Moreover, we utilize $\boldsymbol{W}_1^l$ and $\boldsymbol{W}_2^l$ to represent the weights in the feed-forward layer.

## B.4 Dataset-aware predictor in DiffusionNAG

To ensure that the predictor is dataset-aware, we extend its design by incorporating not only the architecture encoder but also a dataset encoder. In this section, we provide a detailed explanation of this dataset-aware predictor configuration utilized in Section 3.1 of the main paper.

**Neural Architecture Encoding**    For encoding neural architectures, we employ the DiGCN layer (Wen et al., 2020), which is a modified version of Graph Convolutional Networks (GCNs) specifically designed for directed graphs. This layer is specifically designed to effectively capture structural information and dependencies within the architecture. To encode the architecture $\boldsymbol{A}$, represented as $(\boldsymbol{\mathcal{V}}, \boldsymbol{\mathcal{E}}) \in \mathbb{R}^{N \times F} \times \mathbb{R}^{N \times N}$, we first obtain the normalized adjacency matrix $\hat{\boldsymbol{\mathcal{E}}}$ by adding an identity matrix (representing the self-cycles) to the original matrix $\boldsymbol{\mathcal{E}}$ and normalizing it. Subsequently, to enable bidirectional information flow, we utilize two GCN layers. One GCN layer utilizes the normalized adjacency matrix $\hat{\boldsymbol{\mathcal{E}}}$ to propagate information in the "forward" direction, while the other GCN layer uses $\hat{\boldsymbol{\mathcal{E}}}^\top$ to propagate information in the "reverse" direction. The outputs from these two GCN layers are then averaged. In the $l$-th layer of the architecture encoding module, the computation is performed as follows:

$$\boldsymbol{H}^l = \frac{1}{2} \operatorname{ReLU}\left(\hat{\boldsymbol{\mathcal{E}}} \boldsymbol{H}^{l-1} \boldsymbol{W}_+^{l-1}\right) + \frac{1}{2} \operatorname{ReLU}\left(\hat{\boldsymbol{\mathcal{E}}}^\top \boldsymbol{H}^{l-1} \boldsymbol{W}_-^{l-1}\right), \tag{20}$$

where $\boldsymbol{H}^0 = \boldsymbol{\mathcal{V}}$ and $\boldsymbol{W}_+^{l-1}$ and $\boldsymbol{W}_-^{l-1}$ are trainable weight matrices. By stacking multiple layers, we can effectively learn high-quality representations of directed graphs. After applying the graph convolutional layers, we utilize average pooling on the architecture representations obtained from the final layer to aggregate information across the graph. Finally, one or more fully connected layers can be attached to obtain the latent vector $\boldsymbol{z_A}$ of the neural architecture $\boldsymbol{A}$.

**Dataset Encoding**    Following Lee et al. (2021a), we employ a dataset encoder based on Set Transformer (Lee et al., 2019) to accurately capture the characteristics of the target dataset. This dataset encoder is specifically designed to process input sets of varying sizes and output a fixed size latent code, denoted as $\boldsymbol{z_D}$, that effectively encapsulates the information within the dataset. The dataset encoder consists of two stacked modules that are permutation-invariant, meaning output of the dataset encoder is always the same regardless order of elements in the input set. Additionally, these modules leverage attention-based learnable parameters. The lower-level module, known as the "intra-class" encoder, captures class prototypes that represent each class information present in the dataset. On the other hand, the higher-level module, referred to as the "inter-class" encoder, considers the relationships between these class prototypes and aggregates them into a latent vector $\boldsymbol{z_D}$. The utilization of this hierarchical dataset encoder enables us to effectively model high-order interactions among the instances within the dataset, thereby allowing us to capture and summarize crucial information about the dataset.

Ultimately, the encoded representation of the dataset $\boldsymbol{z_D}$ is combined with the encoded architecture representation $\boldsymbol{z_A}$ through MLP layers. This integration enables the predictor to make conditional predictions by considering both the architecture and the dataset information.

## B.5    Transferable Task-guided Neural Architecture Generation

We aim to design a diffusion-based transferable NAG method that can generate optimal neural architectures for **unseen datasets**. In order to achieve this, the surrogate model needs to be conditioned on a dataset $\boldsymbol{D}$ (a.k.a. "dataset-aware") (Lee et al., 2021a; Shala et al., 2023). For the surrogate model to be dataset-aware, we train it over the task distribution $p(\mathcal{T})$ utilizing a meta-dataset $\mathcal{S}$ consisting of (dataset, architecture, accuracy) triplets. To be precise, we define the meta-dataset as $\mathcal{S} \coloneqq \{(\boldsymbol{A}^{(i)}, y_i, \boldsymbol{D}_i)\}_{i=1}^K$ consisting of $K$ tasks, where each task is created by randomly sampling architecture $\boldsymbol{A}^{(i)}$ along with their corresponding accuracy $y_i$ on the dataset $\boldsymbol{D}_i$. We train the dataset-aware accuracy surrogate model using $\mathcal{S}$ by minimizing the following MSE loss function, $\mathcal{L}$:

$$\phi^* \in \arg\min_\phi \sum_{i=1}^K \mathcal{L}(y_i, f_\phi(\boldsymbol{D}_i, \boldsymbol{A}^{(i)})). \tag{21}$$

An important aspect is that this surrogate model only requires a one-time training phase since it is trained over the task distribution. Thereby, the surrogate model can make accurate predictions even for unseen datasets. After one-time training, we utilize this meta-learned **dataset-aware** accuracy surrogate model $f_{\phi^*}(\boldsymbol{D}, \boldsymbol{A}_t)$ as a guiding surrogate. To be precise, we integrate the dataset-aware surrogate model $f_{\phi^*}(\boldsymbol{D}, \boldsymbol{A}_t)$ trained with the objective function (Equation (21)) into the conditional generative process (Equation (9)), which enables us to control the neural architecture generation process for an unseen task, including unseen datasets with the following generation process:

$$\mathrm{d}\boldsymbol{A}_t = \left\{\mathbf{f}_t(\boldsymbol{A}_t) - g_t^2\Big[\boldsymbol{s}_{\boldsymbol{\theta}^*}(\boldsymbol{A}_t, t) + k_t \nabla_{\boldsymbol{A}_t} \log f_{\phi^*}(y|\boldsymbol{D}, \boldsymbol{A}_t)\Big]\right\}\mathrm{d}\bar{t} + g_t \mathrm{d}\bar{\mathbf{w}}. \tag{22}$$

Table 5: **Statistics of NAS-Bench-201 and MobileNetV3 Search Spaces** used in the experiments.

| Search Space | Number of Training Architectures | Number of nodes | Number of operation types |
|---|---|---|---|
| NAS-Bench-201 | 15,625 | 8 | 7 |
| MobileNetV3 | 500,000 | 20 | 9 |

## C   EXPERIMENTAL DETAILS

### C.1   SEARCH SPACE

As explained in Section 2 of the main paper, the architecture $A$ consisting of $N$ nodes is defined by its operator type matrix $\mathcal{V} \in \mathbb{R}^{N \times F}$ and upper triangular adjacency matrix $\mathcal{E} \in \mathbb{R}^{N \times N}$. Thus, $A$ can be represented as $(\mathcal{V}, \mathcal{E}) \in \mathbb{R}^{N \times F} \times \mathbb{R}^{N \times N}$, where $F$ denotes the number of predefined operator sets. Detailed statistics, including the number of nodes ($N$) and the number of operation types ($F$) for each search space utilized in our experiments (excluding Appendix D.3), are provided in Table 5. In Appendix D.3, the only difference is that we train the score network using a subset of 7,812 architectures randomly selected from the NAS-Bench-201 search space. This subset is selected due to utilizing the *Novelty* metric as part of our evaluation process. Further elaboration on the search space is presented below.

**NAS-Bench-201** search space consists of cell-based neural architectures represented by directed acyclic graphs (DAGs). Each cell in this search space originally consists of 4 nodes and 6 edges. The edges offer 5 operation candidates, including `zeroize`, `skip connection`, `1-by-1 convolution`, `3-by-3 convolution`, and `3-by-3 average pooling`. Consequently, there are a total of 15,626 possible architectures within this search space. To utilize the NAS-Bench-201 search space, we convert the original cell representation into a new graph representation. In the new graph, each node represents an operation (layer), and the edges represent layer connections. After the conversion, each cell contains eight nodes and seven operation types for each node. An additional `input` and `output` operation are added to the existing set of 5 operations. To represent the neural architectures in the NAS-Bench-201 search space, we adopt an 8×7 operator type matrix ($\mathcal{V}$) and an 8×8 upper triangle adjacency matrix ($\mathcal{E}$). Furthermore, the macro skeleton is constructed by stacking various components, including one stem cell, three stages consisting of 5 repeated cells each, residual blocks (He et al., 2016) positioned between the stages, and a final classification layer. The final classification layer consists of an average pooling layer and a fully connected layer with a softmax function. The stem cell, which serves as the initial building block, comprises a `3-by-3 convolution` with 16 output channels, followed by a batch normalization layer. Each cell within the three stages has a varying number of output channels: 16, 32, and 64, respectively. The intermediate residual blocks feature convolution layers with a stride of 2, enabling down-sampling.

**MobileNetV3** search space is designed as a layer-wise space, where each building block incorporates `MBConvs`, `squeeze and excitation` Hu et al. (2018), and `modified swish nonlinearity` to create a more efficient neural network. The search space consists of 5 stages, and within each stage, the number of building blocks varies from 2 to 4. As a result, the maximum number of layers possible in MobileNetV3 is calculated as $5 \times 4 = 20$. For each building block, the kernel size can be chosen from the set $\{3, 5, 7\}$, and the expansion ratio can be chosen from $\{3, 4, 6\}$. This implies that there are a total of $3 \times 3 = 9$ possible operation types available for each layer. To represent the neural architectures in the MobileNetV3 search space, we utilize a 20×9 operator type matrix ($\mathcal{V}$) and a 20×20 upper triangle adjacency matrix ($\mathcal{E}$). Furthermore, the search space contains approximately $10^{19}$ architectures, reflecting the extensive range of choices available.

### C.2   BASELINES

In this section, we provide the description of various Neural Architecture Search (NAS) baselines in Section 3.1 and of the main paper.

**Basic approach**   We begin by discussing the basic NAS baseline approaches. **ResNet** (He et al., 2016) is a popular deep learning architecture commonly used in computer vision tasks. Following Lee et al. (2021a), we report results obtained from ResNet56. Random Search (**RS**, Bergstra & Bengio (2012)) is a basic NAS approach that involves randomly sampling architectures from the search space and selecting the top-performing one. Another basic NAS method we consider is **REA** (Real et al., 2019). REA utilizes evolutionary aging with tournament selection as its search strategy. **REINFORCE** (Williams, 1992) is a reinforcement learning based NAS method. In this approach, the validation accuracy obtained after 12 training epochs is used as the reward signal.

**One-shot NAS**   We also compare our method with various one-shot methods, which have shown strong empirical performance for NAS. **SETN** (Dong & Yang, 2019a) focuses on selectively sampling competitive child candidates by training a model to evaluate the quality of these candidates based on their validation loss.

Table 6: **Search Time of One-shot NAS.**

| Method | CIFAR-10 | | CIFAR-100 | | Aircraft | | Oxford-IIIT Pets | |
|---|---|---|---|---|---|---|---|---|
| | Accuracy (%) | GPU Time (s) | Accuracy (%) | GPU Time (s) | Accuracy (%) | GPU Time (s) | Accuracy (%) | GPU Time (s) |
| RSPS (Li & Talwalkar, 2019) | $84.07_{\pm 3.61}$ | 10200 | $52.31_{\pm 5.77}$ | 18841 | $42.19_{\pm 3.88}$ | 18697 | $22.91_{\pm 1.65}$ | 3360 |
| SETN (Dong & Yang, 2019a) | $87.64_{\pm 0.00}$ | 30200 | $59.09_{\pm 0.24}$ | 58808 | $44.84_{\pm 3.96}$ | 18564 | $25.17_{\pm 1.68}$ | 8625 |
| GDAS (Dong & Yang, 2019b) | $93.61_{\pm 0.09}$ | 25077 | $70.70_{\pm 0.30}$ | 51580 | $53.52_{\pm 0.48}$ | 18508 | $24.02_{\pm 2.75}$ | 6965 |
| PC-DARTS (Xu et al., 2020) | $93.66_{\pm 0.17}$ | 10395 | $66.64_{\pm 2.34}$ | 19951 | $26.33_{\pm 3.40}$ | 3524 | $25.31_{\pm 1.38}$ | 2844 |
| DrNAS (Chen et al., 2021) | $94.36_{\pm 0.00}$ | 21760 | $\mathbf{73.51_{\pm 0.00}}$ | 34529 | $46.08_{\pm 7.00}$ | 34529 | $26.73_{\pm 2.61}$ | 6019 |

**GDAS** (Dong & Yang, 2019b) is a differentiable neural architecture sampler that utilizes the Gumbel-Softmax relaxation technique. GDAS sampler is designed to be trainable and optimized based on the validation loss after training the sampled architecture. This allows the sampler to be trained in an end-to-end fashion using gradient descent. **PC-DARTS** (Xu et al., 2020) is a gradient-based NAS method that improves efficiency by partially sampling channels. By reducing redundancy and incorporating edge normalization, PC-DARTS enables more efficient architecture search. **DrNAS** (Chen et al., 2021) is a differentiable architecture search method that formulates the search as a distribution learning problem, using Dirichlet distribution to model the architecture mixing weight. By optimizing the Dirichlet parameters with gradient-based optimization and introducing a progressive learning scheme to reduce memory consumption, DrNAS achieves improved generalization ability.

**Bayesian Optimization based NAS**    A variety of Bayesian Optimization (BO) methods have been proposed for NAS, and in our study, we conduct a comprehensive comparison of these methods. One approach involves using a vanilla Gaussian Process (GP) surrogate (Snoek et al., 2012), where the Upper Confidence Bound (UCB) acquisition function is employed (**GP-UCB**). Additionally, we evaluate our method against other approaches such as **HEBO** (Cowen-Rivers et al., 2022), which is a black-box Hyperparameter Optimization (HPO) method that addresses challenges like heteroscedasticity and non-stationarity through non-linear input and output warping and robust acquisition maximizers. Another BO-based NAS method, **BANANAS** (White et al., 2021a), utilizes an ensemble of fully-connected neural networks as a surrogate and employs path encoding for neural architectures. BANANAS initiates a new architecture search for each task. On the other hand, **NASBOWL** (Ru et al., 2021) combines the Weisfeiler-Lehman graph kernel with a Gaussian process surrogate, allowing efficient exploration of high-dimensional search spaces.

**TransferNAS and TransferNAG**    The key distinction between TransferNAS(NAG) methods and conventional NAS methods lies in their ability to leverage prior knowledge for rapid adaptation to unseen datasets. Unlike conventional NAS methods that start the architecture search from scratch for each new dataset, Transfer-NAS(NAG) methods utilize prior knowledge gained from previous tasks to accelerate the search process on new datasets. By leveraging this prior knowledge, these methods can expedite the search process and potentially yield better performance on unseen datasets.

One of the TransferNAG methods called **MetaD2A** (Lee et al., 2021a), employs an amortized meta-learning approach to stochastically generate architectures from a given dataset within a cross-modal latent space. When applied to a test task, MetaD2A generates 500 candidate architectures by conditioning on the target dataset and then selects the top architectures based on its dataset-aware predictor. **TNAS** (Shala et al., 2023), another TransferNAS method, is a subsequent work of MetaD2A that aims at enhancing the dataset-aware predictor's adaptability to unseen test datasets utilizing BO with the deep-kernel GP. TNAS achieves this by making the deep kernel's output representation be conditioned on both the neural architecture and the characteristics of a dataset. The deep kernel is meta-trained on the same training dataset used in MetaD2A and DiffusionNAG (Ours). In the test phase, TNAS adapts to the test dataset using BO with the deep-kernel GP strategy. As described in Shala et al. (2023), TNAS starts the BO loop by selecting the top-5 architectures with the highest performance on the meta-training set. However, this starting point can be detrimental when searching for an optimal architecture on the unseen target dataset since it may start the BO loop from architectures that perform poorly on the target dataset. As a result, it can lead to a long search time and high computational cost to obtain the target performance on the target dataset.

## C.3    SEARCH TIME OF ONE-SHOT NAS

As described in Table 2 of the main paper, we provide the search time for each dataset of the one-shot NAS methods in Table 6.

## C.4    DATASETS

We evaluate our approach on four datasets following Lee et al. (2021a): CIFAR-10 (Krizhevsky, 2009), CIFAR-100 (Krizhevsky, 2009), Aircraft (Maji et al., 2013), and Oxford IIT Pets (Parkhi et al., 2012), as described in Section 3.2 of the main paper. In this section, we present a detailed description of each dataset used in our

study. **1) CIFAR-10** consists of $32\times32$ color images from 10 general object classes. The training set has 50,000 images (5,000 per class), and the test set has 10,000 images (1,000 per class). **2) CIFAR-100** contains colored images from 100 fine-grained general object classes. Each class has 500 training images and 100 test images. **3) Aircraft** is a fine-grained classification benchmark dataset with 10,000 images from 30 aircraft classes. All images are resized to $32\times32$. **4) Oxford-IIIT Pets** is a dataset for fine-grained classification, including 37 breeds of pets with approximately 200 instances per class. The dataset is divided into 85% for training and 15% for testing. All images are resized to $32\times32$. For CIFAR-10 and CIFAR-100, we use the predefined splits from the NAS-Bench-201 benchmark. For Aircraft and Oxford-IIIT Pets, we create random validation and test splits by dividing the test set into two equal-sized subsets.

### C.5 TRAINING PIPELINE FOR NEURAL ARCHITECTURES

In this section, we provide the architecture training pipeline for both the NAS-Bench-201 and MobileNetV3 search spaces.

**NAS-Bench-201** Following the training pipeline presented in Dong & Yang (2020b), we train each architecture using SGD with Nesterov momentum and employ the cross-entropy loss for 200 epochs. For regularization, we set the weight decay to 0.0005 and decay the learning rate from 0.1 to 0 using a cosine annealing schedule (Loshchilov & Hutter, 2016). We maintain consistency by utilizing the same set of hyperparameters across different datasets. Data augmentation techniques such as random flip with a probability of 0.5, random cropping of a $32\times32$ patch with 4 pixels padding on each border, and normalization over RGB channels are applied to each dataset.

**MobileNetV3** In our training pipeline, we fine-tuned a subnet of a pretrained supernet from the MobileNetV3 search space on the ImageNet 1K dataset for CIFAR-10, CIFAR-100, Aircraft, and Oxford-IIIT Pets datasets, following the experimental setup by Cai et al. (2020). The process involved several steps. Firstly, we activated the subnet and its parameters pretrained on ImageNet 1K for each neural architecture. Then, we randomly initilized the fully-connected layers for the classifier, where the output size of the classifier is matched with the number of classes in the target dataset. Next, we conducted fine-tuning on the target dataset for 20 epochs. We follow the training settings of MetaD2A Lee et al. (2021a) and NSGANetV2 Lu et al. (2020). Specifically, we utilized SGD optimization with cross-entropy loss, setting the weight decay to 0.0005. The learning rate decayed from 0.1 to 0 using a cosine annealing schedule (Loshchilov & Hutter, 2016). We resized images as $224\times224$ pixels and applied dataset augmentations, including random horizontal flip, cutout DeVries & Taylor (2017) with a length of 16, AutoAugment Cubuk et al. (2018), and droppath Huang et al. (2016) with a rate of 0.2. To ensure a fair comparison, we applied the same training pipeline to all architectures obtained by our TransferNAG and the baseline NAS methods.

In Section 3.2 of the main paper, we leveraged a surrogate strategy to report the performance of architectures, which is inspired by the existing surrogate NAS benchmarks Li et al. (2021); Zela et al. (2022); Yan et al. (2021b) that utilize predictors to estimate the performance of architectures within a search space. This approach eliminates the need for exhaustively evaluating all architectures in the large search spaces. For evaluating the performance of each neural architecture, we utilized an accuracy predictor provided by Cai et al. (2020). They randomly sampled 16K subnets from the MobileNetV3 search space with diverse architectures. The accuracy of these subnets was measured on a validation set comprising 10K images randomly selected from the original training set of ImageNet 1K. By training an accuracy predictor using these [architecture, accuracy] pairs, Cai et al. (2020) obtained a model capable of predicting the accuracy of a given architecture. Notably, Cai et al. (2020) demonstrated that the root-mean-square error (RMSE) between the predicted and estimated accuracy on the test set is as low as 0.21%. To ensure a fair comparison, we applied an identical training pipeline to all architectures obtained by our method and the baseline NAS methods.

### C.6 ACQUISITION FUNCTION OPTIMIZATION STRATEGY IN BAYESIAN OPTIMIZATION BASED NAS

In Section 3.2 of the main paper, we address the limitation of existing Bayesian Optimization (BO) based NAS methods by incorporating our conditional architecture generative framework into them. Specifically, we replace the conventional acquisition optimization strategy with our conditional generative framework, guided by the predictor. To facilitate understanding, we present an algorithm for *General Bayesian Optimization NAS* that utilizes an ensemble of neural predictors. Additionally, we demonstrate an algorithm for *Bayesian Optimization NAS with DiffusionNAG*, where our conditional generative framework replaces the conventional acquisition optimization strategy.

---

**Algorithm 1:** General Bayesian Optimization NAS

---

**Input:** Search space $\mathcal{A}$, dataset $\boldsymbol{D}$, parameters $n_0$, $N$, $c$, acquisition function $\varphi$, function
$h : \mathcal{A} \rightarrow \mathbb{R}$ returning validation accuracy of an architecture $\boldsymbol{A} \in \mathcal{A}$.

Draw $n_0$ architectures $\boldsymbol{A}^{(1)}, \ldots, \boldsymbol{A}^{(n_0)}$ uniformly from the search space $\mathcal{A}$ and train them on $\boldsymbol{D}$
to construct $\boldsymbol{B}_{n_0} = \{(\boldsymbol{A}^{(1)}, h(\boldsymbol{A}^{(1)})), \ldots, (\boldsymbol{A}^{(n_0)}, h(\boldsymbol{A}^{(n_0)}))\}$.

**for** $n = n_0, \ldots, N - 1$ **do**

> Train an ensemble of $M$ surrogate model $\{\hat{h}_{\psi_m}\}_{m=1}^M$ based on the current population $\boldsymbol{B}_n$:
>
> $\psi_m^* \in \arg\min_{\psi_m} \sum_{i=1}^n \left( h(\boldsymbol{A}^{(i)}) - \hat{h}_{\psi_m}(\boldsymbol{A}^{(i)}) \right)^2$ for $m = 1, \ldots, M$.
>
> Generate a set of $c$ candidate architectures from $\mathcal{A}$ using an acquisition optimization strategy.
> For each candidate architecture $\boldsymbol{A}$, evaluate the acquisition function $\varphi(\boldsymbol{A})$ with $\{\hat{h}_{\psi_m}\}_{m=1}^M$.
> Select architecture $\boldsymbol{A}^{(n+1)}$ which maximizes $\varphi(\boldsymbol{A})$.
> Train architecture $\boldsymbol{A}^{(n+1)}$ to get the accuracy $h(\boldsymbol{A}^{(n+1)})$ and add it to the population $\boldsymbol{B}_n$:
> $\boldsymbol{B}_{n+1} = \boldsymbol{B}_n \cup \{(\boldsymbol{A}^{(n+1)}, h(\boldsymbol{A}^{(n+1)}))\}$.

**Output:** $\boldsymbol{A}^* = \arg\max_{n=1,\ldots,N} h(\boldsymbol{A}^{(n)})$

---

**Algorithm 2:** Bayesian Optimization with DiffusionNAG

---

**Input:** Pre-trained score network $\boldsymbol{s}_{\boldsymbol{\theta}^*}$, search space $\mathcal{A}$, dataset $\boldsymbol{D}$, parameters $n_0$, $N$, $c$,
acquisition function $\varphi$, function $h : \mathcal{A} \rightarrow \mathbb{R}$ returning validation accuracy of an
architecture $\boldsymbol{A} \in \mathcal{A}$.

Draw $n_0$ architectures $\boldsymbol{A}^{(1)}, \ldots, \boldsymbol{A}^{(n_0)}$ uniformly from the search space $\mathcal{A}$ and train them on $\boldsymbol{D}$
to construct $\boldsymbol{B}_{n_0} = \{(\boldsymbol{A}^{(1)}, h(\boldsymbol{A}^{(1)})), \ldots, (\boldsymbol{A}^{(n_0)}, h(\boldsymbol{A}^{(n_0)}))\}$.

**for** $n = n_0, \ldots, N - 1$ **do**

> Train an ensemble of $M$ surrogate model $\{\hat{h}_{\psi_m}\}_{m=1}^M$ based on the current population $\boldsymbol{B}_n$:
>
> $\psi_m^* \in \arg\min_{\psi_m} \sum_{i=1}^n \left( h(\boldsymbol{A}^{(i)}) - \hat{h}_{\psi_m}(\boldsymbol{A}^{(i)}) \right)^2$ for $m = 1, \ldots, M$.
>
> $\hat{\mu}(\boldsymbol{A}^{(i)}) := \frac{1}{M} \sum_{m=1}^M \hat{h}_{\psi_m^*}(\boldsymbol{A}^{(i)})$
>
> $\hat{\sigma}_i(\boldsymbol{A}^{(i)}) := \sqrt{\frac{\sum_{m=1}^M ((\hat{h}_{\psi_m^*}(\boldsymbol{A}^{(i)}) - \mu(\boldsymbol{A}^{(i)}))^2}{M-1}}$
>
> $p(y|\boldsymbol{A}^{(i)}; \boldsymbol{D}) := \mathcal{N}(y; \hat{\mu}((\boldsymbol{A}^{(i)})), \hat{\sigma}(\boldsymbol{A}^{(i)})^2)$.
>
> /* Generate $c$ candidate architecture with the score network $\boldsymbol{s}_{\boldsymbol{\theta}^*}$. */
> $S \leftarrow \emptyset$
> Minimize $\sum_{i=1}^n -\log p_\xi(y_i|\boldsymbol{A}^{(i)})$ w.r.t $\xi$, where $y_i = h(\boldsymbol{A}^{(i)})$.
> **for** $i = 1, \ldots, c$ **do**
>
> > Draw a random noise $\boldsymbol{A}_T^{(n+i)}$ from the prior distribution.
> > Generate architecture $\boldsymbol{A}^{(n+i)}$ with $\boldsymbol{s}_{\boldsymbol{\theta}^*}$ by the reverse process:
> > $d\boldsymbol{A}_t^{(n+i)} =$
> > $\left\{ \mathbf{f}_t(\boldsymbol{A}_t^{(n+i)}) - g_t^2 \left[ \boldsymbol{s}_{\boldsymbol{\theta}^*}(\boldsymbol{A}_t^{(n+i)}, t) + k_t \nabla_{\boldsymbol{A}_t^{(n+i)}} \log p_\xi(y|\boldsymbol{A}_t^{(n+i)}) \right] \right\} d\bar{t} + g_t d\bar{\mathbf{w}}$
> > $S \leftarrow S \cup \{\boldsymbol{A}_0^{(n+i)}\}$
>
> For each candidate architecture $\boldsymbol{A} \in S$, evaluate $\varphi(\boldsymbol{A})$ with $p(y|\boldsymbol{A}^{(i)}; \boldsymbol{D})$.
> Select architecture $\boldsymbol{A}^{(n+1)}$ which maximizes $\varphi(\boldsymbol{A})$.
> Train architecture $\boldsymbol{A}^{(n+1)}$ to get the accuracy $h(\boldsymbol{A}^{(n+1)})$ and add it to the population $\boldsymbol{B}_n$:
> $\boldsymbol{B}_{n+1} = \boldsymbol{B}_n \cup \{(\boldsymbol{A}^{(n+1)}, h(\boldsymbol{A}^{(n+1)}))\}$.

**Output:** $\boldsymbol{A}^* = \arg\max_{n=1,\ldots,N} h(\boldsymbol{A}^{(n)})$

---

### C.7 IMPLEMETATION DETAILS

In this section, we provide a detailed description of the DiffusionNAG implementation and the hyperparameters used in our experiments.

**Diffusion Process** In Equation (11), which describes the diffusion process in DiffusionNAG, we set the minimum diffusion variance, denoted as $\sigma_{\min}$, to 0.1 and the maximum diffusion variance, denoted as $\sigma_{\max}$, to 5.0. These values determine the range of diffusion variances used during the diffusion process. Additionally, we utilize 1000 diffusion steps. This means that during the diffusion process, we iterate 1000 times to progressively

Table 7: **Hyperparameter Setting of the Score Network in DiffusionNAG on NAS-Bench-201 Search Space.**

| Hyperparameter | Value |
|---|---|
| The number of transformer blocks | 12 |
| The number of heads $n_{\text{head}}$ | 8 |
| Hidden Dimension of feed-forward layers | 128 |
| Hidden Dimension of transformer blocks | 64 |
| Non-linearity function | Swish |
| Dropout rate | 0.1 |
| Learning rate | $2 \times 10^{-5}$ |
| Batch size | 256 |
| Optimizer | Adam |
| $\beta_1$ in Adam optimizer | 0.9 |
| $\beta_2$ in Adam optimizer | 0.999 |
| Warmup period | 1000 |
| Gradient clipping | 1.0 |

Table 8: **Hyperparameter Setting of the Dataset-aware Surrogate Model in DiffusionNAG on NAS-Bench-201 Search Space.**

| Hyperparameter | Value |
|---|---|
| The number of DiGCN layers | 4 |
| Hidden Dimension of DiGCN layers | 144 |
| Hidden Dimension of dataset encoder | 56 |
| The number of instances in each class | 20 |
| Hidden Dimension of MLP layer | 32 |
| Non-linearity function | Swish |
| Dropout rate | 0.1 |
| Learning rate | $1 \times 10^{-3}$ |
| Batch size | 256 |
| Optimizer | Adam |
| $\beta_1$ in Adam optimizer | 0.9 |
| $\beta_2$ in Adam optimizer | 0.999 |
| Warmup period | 1000 |
| Gradient clipping | 1.0 |

generate and update the architectures. Furthermore, we set the sampling epsilon $\epsilon$ to $1 \times 10^{-5}$. This value is used in the sampling step to ensure numerical stability (Song et al., 2021).

**Score Network** To train the score network $s_\theta$ in DiffusionNAG, we employ the set of hyperparameters in Table 7. We conduct a grid search to tune the hyperparameters. Specifically, we tune the number of transformer blocks from the set $\{4, 8, 12\}$, the number of heads from the set $\{4, 8, 12\}$, and the learning rate from the set $\{2 \times 10^{-2}, 2 \times 10^{-3}, 2 \times 10^{-4}, 2 \times 10^{-5}\}$. Additionally, we apply Exponential Moving Average (EMA) to the model parameters, which helps stabilize the training process and improve generalization.

**Dataset-aware predictor** To train the dataset-aware predictor $f_\phi$ in DiffusionNAG, we employ the set of hyperparameters in Table 8. We conduct a grid search to tune the hyperparameters. Specifically, we tune the number of DiGCN layers from the set $\{1, 2, 3, 4\}$, the hidden dimension of DiGCN layers from the set $\{36, 72, 144, 288\}$, the hidden dimension of dataset encoder from the set $\{32, 64, 128, 256, 512\}$, and the learning rate from the set $\{1 \times 10^{-1}, 1 \times 10^{-2}, 1 \times 10^{-3}\}$.

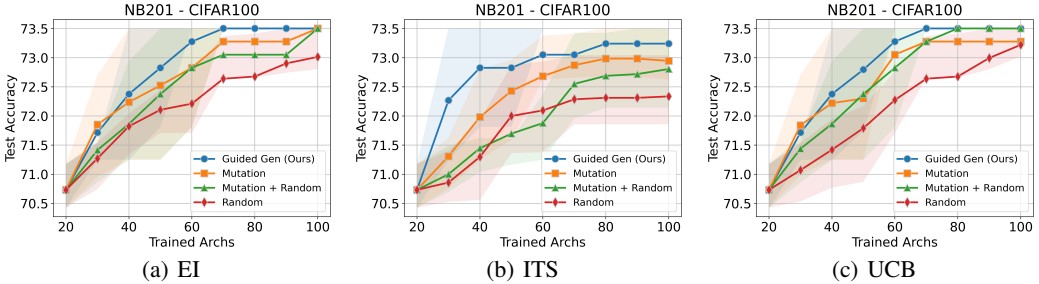

| (a) EI | (b) ITS | (c) UCB |

Figure 6: **Experimental Results on Various Acquisition Functions (NB201).** *Ours* consistently shows superior or comparable performance compared with the heuristic approaches on various acquisition functions. We run experiments with 10 different random seeds.

## D  ADDITIONAL EXPERIMENTAL RESULTS

### D.1  DISTRIBUTION OF GENERATED NEURAL ARCHITECTURES

In Section 3.1 of the main paper, we evaluate the effectiveness of DiffusionNAG in generating high-performing architectures. We generate 1,000 architectures using each method and analyze their distribution. The target datasets ($\boldsymbol{D}$) are CIFAR10/CIFAR100 (Krizhevsky, 2009), for which we have access to the Oracle distribution through the NAS-Bench-201 benchmark (Dong & Yang, 2020b). Alongside the architecture distribution for CIFAR100 depicted in Figure 2 of the main paper, we present the architecture distribution for CIFAR10 in Figure 5. Similar to CIFAR100, DiffusionNAG produces a greater number of high-performing architectures compared to other baselines for the CIFAR10 dataset as well. This observation confirms that the three advantages discussed in Section 3.1 of the main paper also extend to CIFAR10.

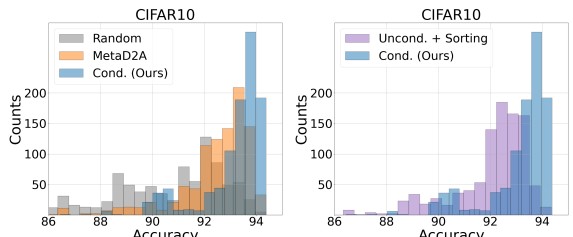

Figure 5: **The Distribution of Generated Architectures for CIFAR10.**

### D.2  IMPROVING EXISTING BAYESIAN OPTIMIZATION NAS ACROSS VARIOUS ACQUISITION FUNCTIONS

Let the actual validation accuracy of an architecture $\boldsymbol{A}$ in the search space $\mathcal{A}$ be denoted as $h(\boldsymbol{A})$, and let $\boldsymbol{B}_n$ represent the set of trained models accumulated during the previous Bayesian Optimization (BO) loops, given by $\boldsymbol{B}_n := \{(\boldsymbol{A}^{(1)}, h(\boldsymbol{A}^{(1)})), \ldots, (\boldsymbol{A}^{(n)}, h(\boldsymbol{A}^{(n)}))\}$. Moreover, we define the mean of predictions as $\hat{\mu}(\boldsymbol{A}) := \frac{1}{M} \sum_{m=1}^{M} \hat{h}_{\psi_m^*}(\boldsymbol{A})$, where $\hat{h}_{\psi_m^*}(\boldsymbol{A})$ represents the prediction made by the $m$-th predictor from an ensemble of $M$ predictors. Additionally, we define the standard deviation of predictions as $\hat{\sigma}(\boldsymbol{A}) := \sqrt{\frac{\sum_{m=1}^{M}((\hat{h}_{\psi_m^*}(\boldsymbol{A}) - \hat{\mu}(\boldsymbol{A}))^2}{M-1}}$. By utilizing an ensemble of predictors $\{\hat{h}_{\psi_m}\}_{m=1}^{M}$, we can estimate the mean and standard deviation of predictions for each architecture in the search space (Please refer to Algorithms 1 and 2 in Appendix C.6).

Following Neiswanger et al. (2019) and White et al. (2021a), we utilize the following estimates of acquisition functions for an input architecture $\boldsymbol{A}$ from the search space $\mathcal{A}$:

$$\varphi_{\text{PI}}(\boldsymbol{A}) = \mathbb{E}\left[\mathbb{1}\left[\hat{\mu}(\boldsymbol{A}) > y_{\max}\right]\right] = \int_{y_{\max}}^{\infty} \mathcal{N}\left(\hat{\mu}(\boldsymbol{A}), \hat{\sigma}^2(\boldsymbol{A})\right) dy \tag{23}$$

$$\varphi_{\text{EI}}(\boldsymbol{A}) = \mathbb{E}\left[\mathbb{1}\left[\hat{\mu}(\boldsymbol{A}) > y_{\max}\right](\hat{\mu}(\boldsymbol{A}) - y_{\max})\right] = \int_{y_{\max}}^{\infty} (\hat{\mu}(\boldsymbol{A}) - y_{\max})\mathcal{N}\left(\hat{\mu}(\boldsymbol{A}), \hat{\sigma}^2(\boldsymbol{A})\right) dy \tag{24}$$

$$\varphi_{\text{ITS}}(\boldsymbol{A}) = \tilde{h}(\boldsymbol{A}), \quad \tilde{h}(\boldsymbol{A}) \sim \mathcal{N}\left(\hat{\mu}(\boldsymbol{A}), \hat{\sigma}^2(\boldsymbol{A})\right) \tag{25}$$

$$\varphi_{\text{UCB}}(\boldsymbol{A}) = \hat{\mu}(\boldsymbol{A}) + \beta\hat{\sigma}(\boldsymbol{A}), \tag{26}$$

where $y_{\max} = h(\boldsymbol{A}^{i^*})$, $i^* = \arg\max_{i \in [n]} h(\boldsymbol{A}^{(i)})$.

We conduct the same experiment as Figure 4 of the main paper in Figure 6 on the different search space, NB201. The experimental results of different acquisition function optimization strategies across various acquisition

functions in NB201 search space are presented in Figure 6. The results demonstrate that our proposed acquisition optimization strategy with the conditional generative framework (**Guided Gen (Ours)**) consistently shows superior or comparable performance compared to conventional acquisition optimization strategies. This improvement is observed across various acquisition functions, including Probability of Improvement (PI), which is used in Section 3.2 of the main paper, as well as Expected Improvement (EI), Independent Thompson sampling (ITS), and Upper Confidence Bound (UCB) in the NB201 search space.

### D.3 Adaptation Across Different Objectives

In Table 9, we demonstrate that DiffusionNAG easily adapts to new tasks by swapping the task-specific predictor in a plug-and-play manner without retraining the score network. We train three predictors with different objectives: **Clean** accuracy, robust accuracy against an adversarial attack such as **APGD** (Croce & Hein, 2020) with a perturbation magnitude of $\epsilon = 2$, and robust accuracy against glass **Blur** corruption. For each objective, we randomly select 50 neural architectures from the NB201 and collect their corresponding test accuracy data on CIFAR10 as training data (i.e., (architecture, corresponding test accuracy) pairs). Subsequently, we train each predictor using their respec-

Table 9: **Adaptation ability across different objectives.**

| | | Stats. | Guidance | | |
|---|---|---|---|---|---|
| | | | $f_{\phi_{\text{Clean}}}$ | $f_{\phi_{\text{APGD}}}$ | $f_{\phi_{\text{Blur}}}$ |
| Test Accuracy | Clean | Max | **93.90** | 80.40 | 86.60 |
| | | Mean | **93.17** | 69.24 | 74.23 |
| | APGD | Max | 2.60 | **26.60** | 22.50 |
| | | Mean | 1.51 | **17.66** | 14.02 |
| | Blur | Max | 37.10 | 53.50 | **56.90** |
| | | Mean | 33.28 | 40.46 | **47.88** |

tive training data, denoted as $f_{\phi_{\text{Clean}}}$, $f_{\phi_{\text{APGD}}}$, and $f_{\phi_{\text{Blur}}}$. With the guidance of these predictors, we generate a pool of 20 architectures for each. For each generated architecture pool, we provide statistics of all objectives (i.e., test accuracies), including robust accuracies under the APGD attack and glass Blur corruption and clean accuracy from the NB201-based robustness benchmark (Jung et al., 2023).

The maximum (max) and mean accuracies of the pool of architectures generated by using the predictor trained on the target objective are higher than those obtained by using predictors trained with other objectives. For example, architectures generated with the guidance of $f_{\phi_{\text{Clean}}}$ have poor robust performance on the **APGD** attack and **Blur** corruption, with a max accuracy of **2.6%** and **37.10%** for each. In contrast, the guidance of $f_{\phi_{\text{APGD}}}$ and $f_{\phi_{\text{Blur}}}$ yield better architectures with a max accuracy of **26.60%** and **56.90%** on the **APGD** attack and **Blur** corruption, respectively, demonstrating the superior adaptation ability of DiffusionNAG across various objectives.

### D.4 Analysis of Generated Neural Architectures across Different Objectives

In the **Adaptation Across Different Objectives** part of Appendix D.3, we demonstrate the capability of our conditional generative framework to easily adapt to new tasks. This adaptability is achieved by seamlessly swapping the predictor, trained with the specific target objective, without the need to retrain the score network.

By leveraging predictors trained with different objectives, such as clean accuracy, robust accuracy against adversarial attacks (e.g., APGD with perturbation magnitude $\epsilon = 2$), and robust accuracy against glass blur corruption (Blur), we generate a pool of 20 neural architectures in the section. In this section, we analyze the architecture pool generated for each objective. Specifically, we examine the occurrence of each operation type for each node in the architecture pool guided by each objective.

Interestingly, each generated architecture pool exhibits distinct characteristics, as illustrated in Figure 7. In Figure 7(a), we observe that the architecture pool guided by the **Clean Accuracy** objective has a higher proportion of `3-by-3 convolution` operation types compared to the other architecture pools. Furthermore, in Figure 7(b), we can observe that the architecture pool guided by the **Robust Accuracy against APGD Attack** objective exhibits distinct characteristics. Specifically, nodes 1, 2, and 3 are dominated by the `zeroize`, `3-by-3 average pooling`, and `skip connection` operation types, while nodes 4, 5, and 6 are predominantly occupied by the `3-by-3 average pooling` operation type. Lastly, in Figure 7(c), depicting the architecture pool guided by the **Robust Accuracy against Glass Blur Corruption** objective, we observe that nodes 1, 2, and 3 are predominantly occupied by the `zeroize` operation type. Based on our analysis, it is intriguing to observe that the characteristics of the generated architectures vary depending on the objective that guides the architecture generation process.

### D.5 Adaptation Across Different Objectives: Corruptions

In Table 9, we evaluate the adaptation abilities of DiffusionNAG to new tasks by swapping the task-specific predictor in a plug-and-play manner without retraining the score network. In the Section, we conduct further experiments to explore a broader range of corruptions introduced not only in glass blur but also in various corruptions in the NB201-based robustness benchmark (Jung et al., 2023). We conduct experiments to find robust neural architectures for 15 diverse corruptions at severity level 2 in CIFAR-10-C. Similar to Table 9, we demonstrated the capabilities of DiffusionNAG to easily adapt to new tasks without retraining the score network. We achieved this by replacing task-specific predictors in a plug-and-play manner. Specifically, we train a clean predictor, $f_{\phi_{\text{clean}}}$), to predict the clean accuracy of neural architectures using randomly sampled pairs of

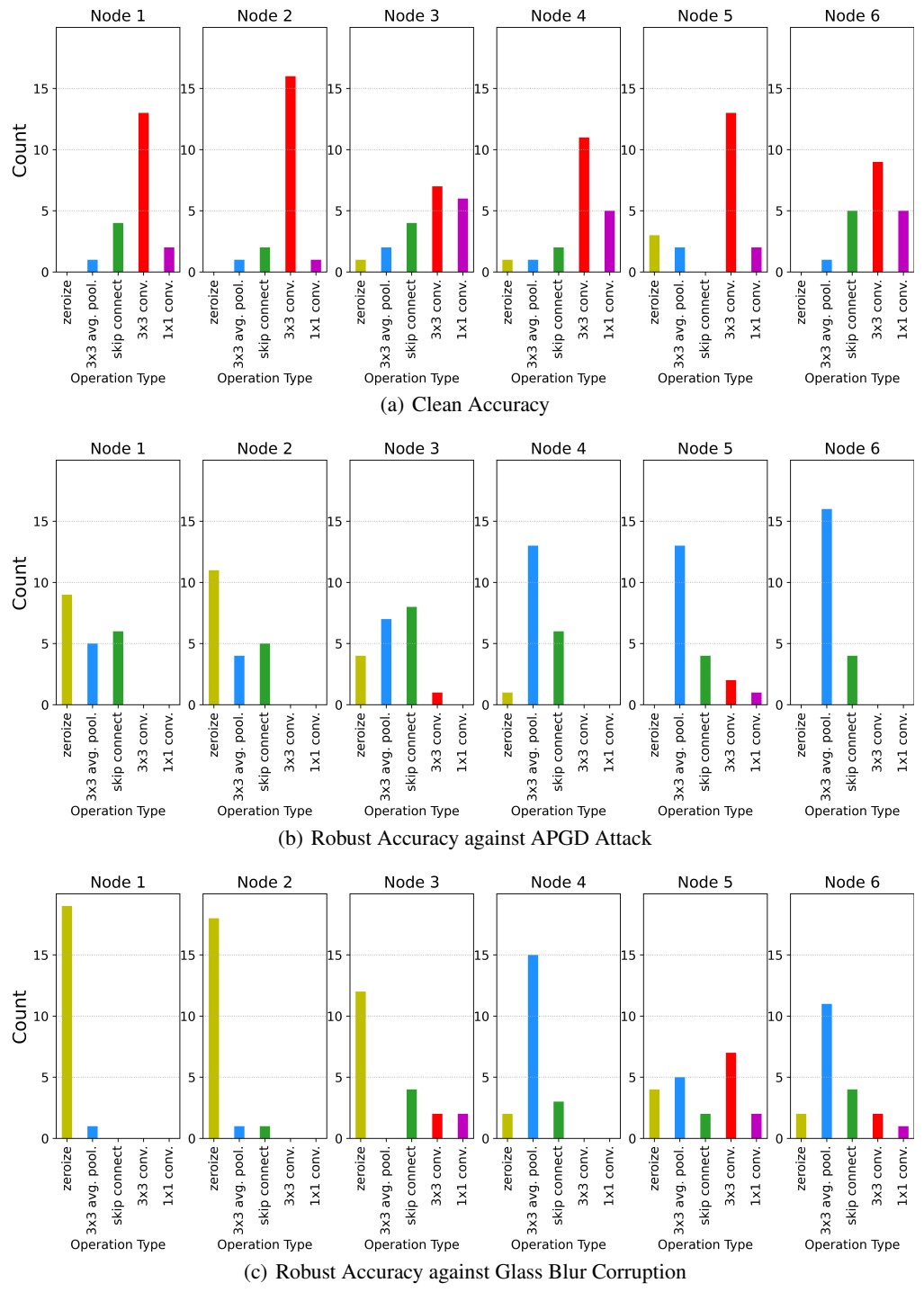

Figure 7: **Analysis of Generated Architectures across Different Objectives.** We analyze the distribution of operation types for each node across the pool of 20 generated architectures. The architectures are generated by DiffusionNAG, guided by surrogate models trained with different objectives such as Clean Accuracy, Robust Accuracy against APGD Attack, and Robust Accuracy against Glass Blur Corruption.

architectures and their corresponding CIFAR-10 clean accuracy from the NB201-based robustness benchmark. Afterward, we train the target corruption predictor, $f_{\phi_{\text{corruption}}}$, to predict the robust accuracy of neural architectures under specific corruption types. We create the training dataset by including pairs of architectures and their respective robust accuracy for each corruption scenario from the NB201-based robustness benchmark. With the guidance of the trained $f_{\phi_{\text{clean}}}$, we generated a pool of 20 architectures ($S_{\text{clean}}$. Similarly, with the guidance of

Table 10: **Adaptation Across Different Objectives: Corruptions.**

| Predictor | Brightness | Contrast | Defocus | Corruption Type Elastic | Fog | Frost | Gaussian | |
|---|---|---|---|---|---|---|---|---|
| $f_{\phi_{\text{clean}}}$ | 89.3 | 58.9 | 64.6 | 61.0 | **74.7** | **60.2** | 30.5 | |
| $f_{\phi_{\text{corruption}}}$ | **89.4** | **59.6** | **64.9** | **61.4** | 74.4 | 58.9 | **49.6** | |
| | Impulse | Jpeg | Motion | Pixelate | Shot | Snow | Zoom | Average |
| $f_{\phi_{\text{clean}}}$ | 55.2 | 61.6 | 43.7 | 71.7 | 39.7 | 70.4 | 44.9 | 59.0 |
| $f_{\phi_{\text{corruption}}}$ | **57.7** | **64.6** | **49.2** | **72.1** | **51.0** | **71.2** | **52.9** | **62.6** |

the task-specific predictors ($f_{\phi_{\text{corruption}}}$), we generate another pool of 20 architectures for each target corruption ($S_{\text{corruption}}$). Subsequently, we retrieve the robust accuracy on both $f_{\phi_{\text{clean}}}$-guided and $f_{\phi_{\text{corruption}}}$-guided architecture pools ($S_{\text{clean}}$ and $S_{\text{corruption}}$) from the NB201-based robustness benchmark for each corruption scenario.

The maximum accuracies of the architecture pool generated using the predictor $f_{\phi_{\text{corruption}}}$ trained for the target corruption surpass those obtained using the accuracy predictor $f_{\phi_{\text{clean}}}$ trained for clean accuracy. For 13 out of the 15 corruptions, excluding fog and frost corruption, the maximum accuracy of the architecture pool guided by the $f_{\phi_{\text{corruption}}}$ predictor exceeds that of the pool guided by $f_{\phi_{\text{clean}}}$. The average maximum accuracy increased by 3.6%, reaching 59.0% and 62.6%, respectively.

