# OpenReview forum: "DiffusionNAG: Predictor-guided Neural Architecture Generation with Diffusion Models"
_ICLR.cc/2024/Conference — ICLR 2024 poster_

### Official Review · Reviewer_EDN5 · 2023-11-01

**Soundness:** 3 good
**Presentation:** 3 good
**Contribution:** 2 fair
**Rating:** 6
**Confidence:** 3

**Summary:**

The paper proposes a method to perform neural architecture search using diffusion models. More precisely, they model the neural architecture as a stochastic process and model it using diffusion. Then at inference with appropriate initialization the reverse process samples a new neural architecture to train a model.

**Strengths:**

1. The use of diffusion models to neural architecture search is in an interesting idea.
2. Modeling NAS as a generative problem enables creating multiple architectures starting with random noise, compared to exhaustively searching the entire space. This enables reducing the NAS cost.
3. Empirically the method outperforms the baselines considered in the paper.

**Weaknesses:**

1. Even though the motivation to the use of diffusion models to neural architecture search is interesting, it seems more appropriate to use discrete diffusion models for discrete random variables instead of using a continuous variables. For instance [1] is an interesting approach which seems to be more correlated to the desired problem instead of continuous diffusion.
2. The use of diffusion models for NAS seems to be very incremental, as the proposed approach is a direct application of the standard available diffusion based approaches. What are some of the major hurdles faced by the authors in making this work for NAS?
3. For conditional NAG, the authors can consider convert A_t to A_0 using the appropriate equations depending on the choice of diffusion sampler used and then apply the pre-trained predictor. This can truly enable using a pre-trained predictor which is just a function of A_0 instead of A_t.

[1]Vector Quantized Diffusion Model for Text-to-Image Synthesis(https://arxiv.org/abs/2111.14822)

**Questions:**

Can such an approach be used to find optimal architectures for diffusion models? Instead of the standard classification problems.
One can start with discrete NAR transformers for instance (like MaskGIT) which can model the NAS process and token predictions, and then the produced model will be used for generation which is another variant of cross entropy minimization problem. Since the foundation models are increasing in size day by day it will be interesting to see if their training methods can be used to find optimal sparse architectures.

[1]MaskGIT: Masked Generative Image Transformer (https://arxiv.org/abs/2202.04200)

---

> ### Author Response · Authors · 2023-11-16
> **Rebuttal by Authors (1/5)**
>
> **W1. Even though the motivation to the use of diffusion models to NAS is interesting, using discrete diffusion models seems more appropriate.**
>
> - Thank you for mentioning that our motivation for applying the diffusion model to NAS is interesting. Additionally, we appreciate your insightful suggestions regarding the types of diffusion models to address NAS challenges. The most critical aspect of applying the diffusion model to NAS is the ability to generate a **valid** neural architecture (a valid graph) while adhering to the rules of the given search space. Therefore, we believe that **as long as we can generate valid neural architectures, we can utilize both continuous and discrete diffusion models**. As suggested by the reviewer, we agree that discrete diffusion models also hold promise in neural architecture generation, as demonstrated by their excellent performance in image synthesis [5].
>
> - In our case, by referring to recent research works [1, 2, 3, 4] that highlight the superior performance of continuous graph diffusion models in generating undirected graphs, we have adopted continuous diffusion models as our primary framework. Following those prior works, we treat graphs with various node/edge categories as entities embedded in continuous space. The node/edge matrix is represented as a binary matrix through the one-hot encoding of features. We discretize the entries of the architecture matrices using the threshold $0.5$ to obtain discrete 0-1 matrices after generating samples by simulating the reverse diffusion process (please see **Section 2**, the reverse process of the main paper for a more detailed explanation).
>
> - Furthermore, please note that, due to the unique characteristics of neural architectures and the diverse search spaces in NAS tasks, merely applying existing diffusion models to NAS without careful considering these characteristics might result in the failure to generate **valid** neural architectures. To address this challenge, we have developed a novel score network for neural architectures to effectively capture the data distribution of neural architectures within the given search space. As shown in **Table 4** of the main paper, **our proposed diffusion model successfully achieved 100.00% validity in both the NB201 and MBv3 search spaces**, while the existing graph diffusion model, GDSS [2], totally fails as 4.56% and 0.00%, respectively. (For a detailed explanation of the challenges in generating valid neural architectures and our technical contributions in designing the score network, please refer to our response to **W2**.)
>
> - Additionally, in the sections **Q1. The feasibility of applying DiffusionNAG to generate optimal generative models** and **Q2. The feasibility of applying DiffusionNAG to generate sparse foundation models**, we discuss the potential for future research utilizing the discrete diffusion models.
>
> ---
> **References**
>
> [1] Permutation invariant graph generation via score-based generative modeling, ICML 2020.
>
> [2] Score-based Generative Modeling of Graphs via the System of Stochastic Differential Equations, ICML 2022.
>
> [3] Exploring Chemical Space with Score-based Out-of-distribution Generation, ICML 2023.
>
> [4] Graph Generation with Destination-Predicting Diffusion Mixture, Arxiv 2023.
>
> [5] Vector quantized diffusion model for text-to-image synthesis, CVPR 2022.

---

> ### Author Response · Authors · 2023-11-16
> **Rebuttal by Authors (2/5)**
>
> **W2. Distinguishing features from other graph diffusion models.**
>
> **[Technical novelty of DiffusionNAG as a diffusion model for NAS]**
> - The graph tasks typically handled by existing diffusion models, such as molecules or community connections, are undirected, and the order of node generation is relatively less critical. However, when dealing with neural architectures, the graph structures become more intricate due to 1) the necessity of reflecting the computational flow from the input layer to the output layer and 2) the requirement to adhere to complex and specific rules that vary across different search spaces (Please refer to the **[Validity criteria]** section below for more detailed explanations regarding validity rules). Therefore, to generate valid graphs representing neural architectures, diffusion models should capture 1) the **dependency between nodes**, reflecting the computational flow, and 2) the **accurate position of each layer** within the overall neural architecture to comply with the rules of a specific search space.
>
> - However, to the best of our knowledge, none of the existing graph diffusion models can effectively capture the directionality of graphs or provide a specialized representation to encode the accurate flow of topological ordering. Consequently, as illustrated in Table 4 of our main paper, directly applying the existing diffusion-based models, such as **GDSS**, leads to difficulties in generating valid architectures. The validity ratios are only **4.56%** and **0.00%** for the NB201 and MBv3 search spaces, respectively.
>
> - To address this issue, we first represented neural architectures as **Directed Acyclic Graphs (DAGs) with an upper triangular adjacency matrix**. Then we designed a novel score network to generate valid neural architectures. As detailed in **Section 2** of the main paper, we employed **transformer blocks** with an attention mask that indicates the dependency between nodes to parameterize the score network. Finally, we incorporated **positional embeddings** for nodes, as shown in **Eq (4)**, to capture the accurate topological ordering of layers in architectures.
>
> - As shown in **Table 4** of the main paper, our DAG representation and transformer-based score network (without positional embeddings) to capture the computational flow (node dependency) of neural architectures achieved **42.17%** validity in MBv3 search space while the existing graph diffusion model, **GDSS**, totally failed as **0.0%**. Notably, our final score network with positional embedding achieved **100.0% validity**, highlighting the necessity of positional embedding for generating architectures to follow complex rules of MBv3 search space.
>
> **[Validity criteria]**
>
> - NB201 search space:
>   - 1) Single operation per layer: Each layer should correspond to only one type of operation, avoiding the presence of multiple operations for a single layer.
>   - 2) Capturing the start and end of the computational flow: In the cell-based NB201 search space, the initial node where the computational flow begins must have the operation corresponding to START, and the final node where the computational flow concludes should have the END operation. Additionally, there should be no operations corresponding to START and END between the initial and final nodes.
> - MBv3 search space:
>   - 1) Single operation per layer: Each layer should correspond to only one type of operation, avoiding the presence of multiple operations for a single layer.
>   - 2) No empty stage: The MBv3 search space defines a fixed macro skeleton with five stages. Each stage must consist of at least two layers to prevent the occurrence of an empty (blocked) stage. This configuration, with a directed edge from the node immediately before it, ensures a continuous main path connecting all nodes.
>   - 3) No isolated paths: Within each stage, no layer should lack predecessors, ensuring that there are no isolated paths.
>
> **[Contributions to NAS field]**
> - The proposed DiffusionNAG not only introduces a score network for NAS but also makes significant contributions by addressing key limitations in conventional NAS methods, offering efficient search, superior predictor utilization, and easy adaptability across tasks. Specifically, we pioneer a paradigm shift towards a Conditional Neural Architecture Generation (NAG) scheme from NAS, enhancing search efficiency by generating neural architectures aligned with a specific distribution. Furthermore, DiffusionNAG extends the predictor's role for both NAG and evaluation purposes, while the traditional approaches restrict the same predictor to the evaluation role. Additionally, DiffusionNAG can generate architectures for unseen tasks, thereby improving the generalization capability compared with most NAS methods. Through extensive experiments in two critical scenarios, Transferable NAS and BO-NAS, DiffusionNAG consistently demonstrates these contributions successfully by outperforming existing NAS methods.

---

> ### Author Response · Authors · 2023-11-16
> **Rebuttal by Authors (3/5)**
>
> **W3. A suggestion to use the predictor with input $A_0$ instead of $A_t$.**
>
> - Thank you for suggesting an insightful perspective. As you mentioned, if we can accurately transform $A_t$ to $A_0$ during the architecture sampling process using the equation related to the diffusion sampler we adopted, we can effectively guide the generation of neural architectures using a predictor that takes $A_0$ as input. However, even if we have already defined the forward diffusion process that perturbs $A_0$ to $A_t$ from the diffusion sampler we adopted, due to the stochastic nature of the diffusion process, it is challenging to accurately predict $A_0$ from given $A_t$ during the sampling of neural architectures which simulates the *reverse* diffusion process described in **Eq (2)**.
> - In more detail, as shown in **Appendix Section B.2**, we utilize the SDE equation to model the forward diffusion process for our neural architecture generation framework. Given the SDE describes the process (please refer to the equation below),  we can only know the mean and standard deviation of the neural architecture distribution at each time step, $p(A_t)$. Thus, it is difficult to accurately predict $A_0$ for a given $A_t$ during the architecture sampling process simulating the reverse diffusion process, which starts from noise sampled from the known prior distribution (please refer to **Eq. (11)** and **Eq. (12)** of the **Appendix** in the main paper for a more detailed description).
>
> $$p_{t|0}(A_{t}|A_0) = \mathcal{N}\left(A_t;  A_{0} \, \sigma_{\min}^2\left(\frac{\sigma_{\max}}{\sigma_{\min}}\right)^{2t}\mathbf{I}\right)$$
>
> **[Empirical observation]**
>
> - Therefore, compared to generating neural architectures using a predictor with $A_t$ as input, it will be less effective to guide neural architecture generation using an appropriate equation depending on the chosen diffusion sampler and a predictor with $A_0$ as input. We have empirically observed this assumption. We compared the results of conditional generation using a predictor with $A_0$ as input alongside the appropriate equation of the chosen diffusion process, with the results obtained using a predictor with the $A_t$ as input (original version). We conducted experiments on the CIFAR-10 dataset within the NB201 search space under the Transferable NAS scenario and generated 100 neural architectures using a predictor with input $A_t$ and a predictor with input $A_0$, respectively. The statistics analyzed from the generated neural architectures are provided in **Table R1**. As shown in **Table R1**, when guiding neural architecture generation using the predictor with input $A_0$, the max accuracy is **93.8633%**, which is lower than the max accuracy of **94.3733%** obtained when guiding with the predictor with input $A_t$. This observation also holds for mean accuracy and min accuracy. Additionally, upon ranking the 100 generated neural architectures based on their predicted accuracies with the predictor, the average accuracy of the Top-5 neural architectures is **94.0440%** when guided by the predictor with input $A_t$, whereas it is significantly lower at **88.4357%** when guided by the predictor with input $A_0$.
>
> **<Table R1>**
> |                                                     | Predictor with input $A_t$ | Predictor with input $A_0$ |
> |:---------------------------------------------------:|:--------------------------:|:--------------------------:|
> |                   Max Accuracy (%)                  |         **94.3733**        |           93.8633          |
> |                  Mean Accuracy (%)                  |         **93.1460**        |           85.5664          |
> |                   Min Accuracy (%)                  |         **88.2650**        |           54.5467          |
> | Average Accuracy of Top-5  sorted architectures (%) |         **94.0440**        |           88.4357          |

---

> ### Author Response · Authors · 2023-11-16
> **Rebuttal by Authors (4/5)**
>
> **Q1. The feasibility of applying DiffusionNAG to generate optimal generative models.**
>
> - Thank you for your interesting suggestion that explores optimal architectures for diffusion models. If we understand your suggestion correctly, you are inquiring about the feasibility of generating optimal generative models using a discrete diffusion model under the proposed DiffusionNAG framework.
> - While there are a lot of design choices to generate diffusion models, let’s consider transformer-based generative models, as you suggested. Inspired by [6], a NAS work focusing on designing vanilla transformers (although not generation model-based NAS work), we can define a search space that encompasses key architectural design choices, such as the number of transformer blocks, the number of MLP layers within each block, the number of attention layers within each block, and the hidden state’s dimension of attention or MLP layers. Similar to a set of visual tokens [7] or vocab tokens in the NLP field, we can create a set of architectural tokens by representing all combinations of architectural design choices as architectural tokens (i.e., one-hot vectors).
> - Similar to dividing a vacant canvas into image patches and predicting visual tokens for each patch [7], in NAS, we can consider that the macro skeleton of architectures (the entire graph) corresponds to the vacant canvas, where nodes of the graph correspond to image patches. Subsequently, we perform architectural token predictions for each node in the graph.
> - To train the score model, our approach involves sampling graphs from the search space, partially masking nodes, and predicting architectural tokens for the masked nodes. This transforms the problem into a cross-entropy minimization task, predicting probabilities for each masked token and computing the cross-entropy between the ground truth one-hot token and the predicted token.
> - The performance metric in this scenario may vary depending on the target generative task. For instance, we can consider FID or IS as performance metrics for image synthesis. We can train the predictor guiding the architecture generation process of DiffusionNAG to predict the performance of an architecture for the target generative task. By utilizing the trained predictor, we can guide the generation process of DiffusionNAG to generate an optimal architecture of a generative model for the target task.
>
> ---
> **References**
>
> [6] HAT: Hardware-Aware Transformers for Efficient Natural Language Processing, ACL 2020.
>
> [7] MaskGIT: Masked Generative Image Transformer, CVPR 2022.

---

> ### Author Response · Authors · 2023-11-16
> **Rebuttal by Authors (5/5)**
>
> **Q2. The feasibility of applying DiffusionNAG to generate sparse foundation models.**
>
> - To explore efficient sparse architectures for foundation models, we can leverage DiffusionNAG, similar to the research idea discussed in our response to **Q1. The feasibility of applying DiffusionNAG to generate optimal generative models**. Among various potential directions, one illustrative example is to consider DiffusionNAG as a post-training structured pruning approach designed to induce sparsity in Large Language Models (LLMs). This post-training structured pruning method starts from pre-trained models and eliminates redundant architecture elements along with their parameters. We find this approach reasonable, particularly when considering the impracticality of iteratively pre-training or iteratively pruning models at the scale of LLMs with hundreds of billions of parameters.
> - Specifically, since the majority of LLMs are transformer-based models, we can explore sparse architectures within the search space that was defined in our response to Q1. Our goal would be to identify the optimal architecture that minimizes the accuracy drops within a resource budget (e.g., memory footprint or inference time). In other words, we aim to find the best architecture that enhances inference speed or reduces memory footprint without compromising accuracy.
> - We can address this problem by employing the proposed DiffusionNAG method. First, we can train the score model of DiffusionNAG in a manner similar to that explained in Q1. Secondly, we can formulate a multi-objective function to evaluate the performance of sparse architecture candidates. For example, we may consider existing multi-objectives proposed by ProxylessNAS [8] or MNASNet [9] to find an optimal balance between accuracy and latency. Using the multi-objective function, we can train the predictor to predict the performance of given architecture candidates.
> - Guided by the trained predictor, DiffusionNAG can generate an optimal sparse architecture (graph). In the final step, we selectively activate only the architectural components and parameters of LLMs that align with the generated graph. In other words, we prune (deactivate) architectural components such as blocks, MLP layers, and attention layers that do not correspond to the graph. This process results in a resource-efficient model, enhancing the model's sparsity and contributing to improved efficiency in terms of inference latency or memory utilization.
>
> ---
> **References**
>
> [8] ProxylessNAS: Direct Neural Architecture Search on Target Task and Hardware, ICLR 2019.
>
> [9] MnasNet: Platform-Aware Neural Architecture Search for Mobile, CVPR 2019.
>
> ---
> We hope that the discussions and additional experiments in this response further strengthen our argument and address the concerns raised by Reviewer EDN5. Thanks to the constructive comments from the reviewer, we have improved the clarity of the technical contributions of our proposed score network. Furthermore, we are delighted to engage in discussions about the exploration of diverse future research inspired by our approach. Particularly, we appreciate the interesting suggestions: 1) Beyond the existing NAS benchmarks focused on classification models, exploring the research opportunity to generate a generative model using our generative model like self-evolving AI. 2) Investigating the applicability of our work to make the foundational model sparse. We included our discussion regarding the future works in **Section E of the Appendix** in the revision. We are grateful for the time and effort spent reviewing our paper. If there is any additional information we should provide, please let us know.

---

> ### Author Response · Authors · 2023-11-20
> **Gentle Reminder - Dear Reviewer EDN5**
>
> Would you mind reviewing our response and revision? We have faithfully addressed your concerns with detailed descriptions, additional experiments and discussions for the future works. Below is a brief summary of our response.
> - We clarified our major challenges (difficult validity conditions of NAS tasks) and technical contributions of our score network.
> - We provided a discussion regarding the availability of using a discrete diffusion model instead of a continuous diffusion model for designing our DiffusionNAG.
> - We provided discussions for future works, including applying our DiffusionNAG to generate 1) optimal generative models and 2) sparse foundation models (Section E of the Appendix).
>
> We are happy to discussing any points in the rebuttal that require further clarification or addressing any additional questions the reviewer may have about the paper. If your concerns are addressed based on the rebuttal, we hope you update your rating accordingly. We appreciate the reviewer’s insightful, creative suggestions and constructive comments once again.
>
> Best,
>
> DiffusionNAG authors

---

> ### Author Response · Authors · 2023-11-23
> **Gentle reminder - The interactive discussion phase will end in less than 10 hours**
>
> Dear Reviewer EDN5,
>
> We sincerely appreciate your time and effort in reviewing our paper.  **Since we have less than 10 hours before the end of the interactive discussion phase, we politely ask you to check our detailed responses and the revision.** We have diligently responded to all your comments and faithfully reflected them in the revision. To further save your time, we briefly summarize our responses below:
>
> ***
> * **Part 1 (1/5)**
>   * We discussed the feasibility of employing a discrete diffusion model instead of a continuous diffusion model in the design of our DiffusionNAG.
> ***
> * **Part 2 (2/5)**
>   * We clarified our major challenges (difficult validity conditions of NAS tasks),  technical contributions of our score network and updated the revision (Section 2.1 & Section 2.3).
>   * In addition, you can refer [the Reviewer ZVno’s comments](https://openreview.net/forum?id=dyG2oLJYyX&noteId=8puvohv4kY) that our challenges and technical contributions to generate valid neural architectures are very educational to the Reviewer ZVno.
> ***
> * **Part 3 (3/5)**
>   * We conducted experiments using the predictor with input $A_0$ instead of $A_t$ (Table R1) and empirically demonstrated that utilizing $A_t$ as an input to the predictor yields better results than using $A_0$.
> ***
> * **Part 4 (4/5)**
>   * We discussed future work aimed at generating optimal generative models utilizing DiffusionNAG and included this discussion in the revision (Section E of the Appendix).
> ***
> * **Part 5 (5/5)**
>   * We discussed future work aimed at generating sparse foundation models utilizing DiffusionNAG and included this discussion in the revision (Section E of the Appendix).
> ***
> We hope that our responses have addressed your concerns, and we kindly hope you consider updating the rating accordingly. Please let us know if there are any other things that we need to clarify or provide. We sincerely appreciate your insightful and constructive comments once again.
>
> Best,
>
> DiffusionNAG Authors

---

### Official Review · Reviewer_ZVno · 2023-11-01

**Soundness:** 3 good
**Presentation:** 3 good
**Contribution:** 3 good
**Rating:** 6
**Confidence:** 2

**Summary:**

The paper proposes DiffusionNAG, a novel conditional Neural Architecture Generation framework based on diffusion models, which considers the neural architectures as directed graphs. The proposed method is shown to be significantly more efficient than previous NAS schemes. Empirically, the authors demonstrate it can provide speedups up to 20x, while being superior in performance.

I would also like to note that I am no expert in Neural Architecture Search, which could affect my judgments below.

**Strengths:**

* The proposed method, which uses diffusion models to essentially generate the entire search space and then utilizes the gradient of property predictor to guide the generation towards the more potent candidate, is novel to my knowledge.
* Diffusion model sampling is essentially treated as a search algorithm, which enables more aggressive searches that result in faster procedures.
* The proposed framework could utilize different predictors dependent on the tasks, so the base diffusion model only needs to be trained once.
* The paper also explores diffusion models suitable for generating directed acyclic graphs, which could be useful for other fields as well.
* Empirically, the DiffusionNAG is shown to be scalable and effective, providing speedups while improving the performances.
* The paper is well-written and easy to follow for the most parts.

**Weaknesses:**

* I think some parts could be explained a bit more, even if they are from prior works, e.g. the meta-learned dataset-aware predictor in Sec.2.3. This, along with the diffusion background, could be covered in a dedicated background section.

**Questions:**

* I would not say that I have the best time understanding the parts about "Score Network for Neural Architectures": what exactly is being pursued here? if our adjacency matrix is defined to be upper triangular, should it not already be a DAG? What is the goal of the proposed positional embedding? A bit more explanation is welcomed here.

---

> ### Author Response · Authors · 2023-11-16
> **Rebuttal by Authors (1/2)**
>
> Thank you for your insightful questions, and we appreciate the opportunity to provide further clarification on the meta-learned dataset-aware predictor and the proposed score network for neural architectures.
>
> ___
>
> **W1. Explain in more detail about the meta-learned dataset-aware predictor in Sec 2.3.**
>
> We provide further clarification on meta-learned dataset-aware predictor as follows:
>
> **The structure of the meta-learned dataset-aware predictor.**
>
> - To ensure that the predictor is dataset-aware, the meta-learned dataset-aware predictor consists of two main components: **1) the architecture encoder** and **2) the dataset encoder**. In the architecture encoder part, the DiGCN layer [1] is employed to encode the architecture input. The objective of this architecture encoder is to obtain the latent vector, denoted as $z_A$, for a given neural architecture $A$. In the dataset encoder part, following [2], we utilize a dataset encoder based on Set Transformer [3] to precisely capture the characteristics of the target dataset. This dataset encoder is specifically designed to process input sets of varying sizes and output a fixed-size latent code, denoted as $z_D$, that effectively captures the information within the dataset, $D$.
> - Ultimately, the encoded representation of the dataset $z_D$ is combined with the encoded architecture representation $z_A$ through MLP layers. This integration enables the predictor to make conditional predictions by considering both the architecture and the dataset information. For a more detailed explanation, please refer to **Section B.4 of the Appendix.**
>
> **Transferable conditional neural architecture generation with meta-learned dataset-aware predictor.**
>
> - The proposed DiffusionNAG can generate optimal neural architectures for unseen datasets. In order to achieve this, we utilized the dataset-conditioned property predictor (a.k.a. "dataset-aware") with the structure described earlier. To enable the predictor to generalize to unseen datasets, we train it over the task distribution $p(\mathcal{T})$ utilizing a meta-dataset $\mathcal{S}$ consisting of (dataset, architecture, accuracy) triplets. To be precise, we define the meta-dataset as $\mathcal{S} = \lbrace (A^{(i)}, y_{i}, D_i )\rbrace_{i=1}^K $ consisting of  $K$ tasks,  where each task is created by randomly sampling architecture $A^{(i)}$ along with their corresponding accuracy $y_i$ on the dataset $D_i$.
> - We meta-train the dataset-aware predictor, $f_{\phi}(\cdot)$, using $\mathcal{S}$ by minimizing the MSE loss function, $\mathcal{L}$, as follow:
>
> $$
>    {\phi}^* \in \underset{\phi}{\operatorname{argmin}} \sum_{i=1}^K \mathcal{L}(y_i, f_{\phi}(D_i, A^{(i)})) .
> $$
>
> - An important aspect is that the dataset-aware predictor is meta-learning-based that only requires a one-time training phase since it is trained over the task distribution. Thereby, the dataset-aware predictor can make accurate predictions even for unseen datasets.
> After one-time training, we utilize this meta-learned dataset-aware accuracy predictor $f_{{\phi}^*}({D}, {A}_t)$ as a guiding surrogate.
>
> - To be precise, we incorporate the dataset-aware predictor $f_{\phi^*}(D, A_t)$, trained using the objective function $\mathcal{L}$, into the conditional generative process described by Equation (9) in the main paper. This integration allows us to control the neural architecture generation process for an unseen task (unseen datasets), as outlined in the following generation process:
>
> $$
> \mathrm{d} A_t = \lbrace \mathbf{f_{t}} (A_t) - {{g_t}^2} \Big[ s_{\theta^*} (A_t, t) + k_t {\nabla_{A_t} \log f_{\phi^{*}} (y|D, A_t)}\Big] \rbrace \mathrm{d}\bar{t} + g_{t}\mathrm{d}\bar{\mathbf{w}}.
> $$
>
> ---
> **References**
>
> [1] Neural predictor for neural architecture search, ECCV 2020.
>
> [2] Rapid neural architecture search by learning to generate graphs from datasets, ICLR 2021.
>
> [3] Set transformer: A framework for attention-based permutation-invariant neural networks. ICML 2019.
>
> ---
>
> Following your comments, we have updated the descriptions of the meta-learned dataset-aware prediction in **Section B.5 of the Appendix**. Thank you once again for your valuable feedback.

---

> ### Author Response · Authors · 2023-11-16
> **Rebuttal by Authors (2/2)**
>
> **Q. Explain in more detail about the score network for neural architectures.**
>
> - To elaborate, our primary objective is to generate **valid** neural architectures. While an upper triangular adjacency matrix representing a DAG helps in defining a neural architecture, it alone falls short in capturing the intricate data distribution of neural architectures within the diverse search spaces inherent to NAS tasks. The complexity arises due to the need to adhere to specific rules that vary across different search spaces, as outlined in the **[Validity Criteria]** section, as well as the computational flow from input to output layers.
>
> - **[Validity Criteria]** For generated neural architectures to be valid within the MBv3 search space, the following rules must be satisfied:
>   - 1) Single operation per layer: Each layer should correspond to only one type of operation, avoiding the presence of multiple operations for a single layer.
>   - 2) No empty stage: The MBv3 search space defines a fixed macro skeleton with five stages. Each stage must consist of at least two layers to prevent the occurrence of an empty (blocked) stage. This configuration, with a directed edge from the node immediately before it, ensures a continuous main path connecting all nodes.
>   - 3) No isolated paths: Within each stage, no layer should lack predecessors, ensuring that there are no isolated paths.
>
> - Therefore, to generate valid graphs representing neural architectures, the score network should capture 1) the dependency between nodes, reflecting the computational flow, and 2) the accurate position of each layer within the overall neural architecture to comply with the rules of a specific search space. **While a DAG representation using an upper triangular matrix might capture the former condition, it might fail to encode the accurate position of each layer to follow complicated rules.**
>
> - To address these challenges, as detailed in Section 2 of the main paper, we have designed a novel score network, which employs transformer blocks and incorporates an upper triangular matrix describing node dependencies within a neural architecture. Moreover, we have introduced **positional embeddings (as depicted in Eq (4)) to more accurately capture the topological ordering of layers in architectures**, which leads to the generation of valid architectures adhering to specific rules within the given search space.
>
> - Empirically, as shown in Table 4 in the main paper, we encountered difficulties in generating valid neural architectures when positional embeddings were absent in the MBv3 search space (**42.17% validity**). However, upon incorporating positional embeddings to capture the topological ordering of neural architectures, we observed a **100% validity** in the MBv3 search space. These experimental results show that positional embeddings are necessary to effectively capture the data distribution of neural architectures within the given search space.
>
> ---
>
> If you have any further questions or if there is anything else you would like clarification on, please feel free to let us know. We are more than happy to provide additional explanations. Thank you for your time and effort in reviewing our paper.

---

> ### Author Response · Authors · 2023-11-20
> **Gentle reminder - Dear Reviewer ZVno**
>
> The interactive discussion phase will end this Wednesday (22nd), and we cannot have discussions with you anymore after the deadline. Would  you please review our response and revision? We have faithfully addressed your concerns by updating descriptions regarding the meta-learned dataset-aware predictor and our score network. We would like to gently remind you about it as follows:
> - We improved clarity by explaining details of prior works, specifically the meta-learned dataset-aware predictor (Section B.5 of the Appendix).
> - We improved clarity by explaining the details of our score network.
>
> We believe that they helped significantly strengthen our paper and if your concerns are addressed based on the rebuttal, we hope you update your rating accordingly. Please let us know if you have any further questions or concerns. We appreciate your constructive comments once again.
>
> Best,
>
> DiffusionNAG authors

---

> ### Comment · Reviewer_ZVno · 2023-11-22
>
> I thank the authors for their effort in providing clarifications, and indeed it has been very educational for me, especially the valid network part. I think it would be great to see them incorporated into the main text (I understand that authors are only having them in the appendix for the time being). I do not have more questions and maintain my suggestion for acceptance.

---

> ### Author Response · Authors · 2023-11-23
> **Thank you for your time and effort in reviewing our paper**
>
> We sincerely appreciate your time and effort in reviewing our paper, as well as your suggestion to incorporate clarifications into the main text. **We are happy to hear that our responses addressed all of your questions**, and we are especially glad to know that you found the discussion on the score network for valid neural architectures to be educational.
>
> Following your suggestion, we have made additional changes in the main text of the new revision (highlighted in blue) as follows:
> * **Section 2.1**: We have provided further details regarding the goal of positional embedding within our proposed score network.
> * **Section 2.3**: We have expanded on the details of the meta-learned dataset-aware predictor by offering more comprehensive explanations of the loss function and meta-dataset.
>
> We believe that your valuable feedback helped significantly strengthen our paper. Thank you for reviewing our paper once again.
>
> Best,
>
> DiffusionNAG Authors

---

### Official Review · Reviewer_qPhs · 2023-11-01

**Soundness:** 3 good
**Presentation:** 3 good
**Contribution:** 3 good
**Rating:** 6
**Confidence:** 3

**Summary:**

The authors propose a diffusion based model for neural architecture generation. By using the guidance from the predictor, DiffusionNAG can generate task-specific architectures. They verify the effectiveness of DiffusionNAG on Transferable NAS and Bayesian Optimization-based NAS.

**Strengths:**

This paper is well-motivated. Previous work only uses the diffusion model to model undirected graphs for neural architecture generation. Using the diffusion model for modeling the directed graph is needed. Moreover, classifier guidance is well suited for the Transferable NAS and substitute step 3) in BO-based NAS approaches, where the dataset and the pre-trained classifier are given as the setup.

**Weaknesses:**

1."Whether diffusion model for neural architecture generation is more efficient than Mutation + Random" is a good question. It is important to ask "when they are better".

* The strength of the predictor guidance is an important hyperparameter. In this paper, a good performance can be achieved by cross-validation. Could the author provide some ablation studies of how this hyperparameter affects the performance? How could we effectively define the search range for cross-validation?

* Will using the diffusion model for neural architecture generation have some model collapse phenomenon? Does the Diffusion model only generate some similar architecture, and achieve the better "worse case accuracy" by sacrificing the diversity? If that is the case, any explore and exploit framework for it?

**Questions:**

Please see the Weaknesses part.

---

> ### Author Response · Authors · 2023-11-16
> **Rebuttal by Authors (1/3)**
>
> Thank you for your thoughtful suggestions. Below, we have addressed your concerns and provided a more detailed explanation of how we define a search range for guidance strength.
>
> **W1-1. Defining the search range for guidance strength.**
>
> To effectively define the search range for guidance strength, we employ the following strategy:
> - **Coarse grid search**: Inspired by [1], we first perform a coarse grid search on the guidance strength within a log scale, identifying a strength where validity becomes stable. For instance, within the NB201 search space under the Transferable NAS scenario (please refer to Table R1), a guidance strength of $10^4$ yielded 100.0% valid neural architectures.
>
> **<Table R1>** Validity results - coarse grid search.
> | Guidance Strength | $10^6$ | $10^5$ | **$10^4$** |
> |:-----------------:|:------:|:------:|:----------:|
> |    Validity (%)   |  0.00  |  0.00  | **100.00** |
>
>
> - **Fine-grained grid search**: Building upon the coarse grid search, we perform a more fine-grained grid search around the point identified in the coarse grid search phase. This involves incrementally adjusting the guidance strength until we identify the optimal point where validity remains intact.
>
> - Subsequently, we employ a **balanced exploitation-exploration sampling strategy** that iteratively decreases the guidance strength from this identified point through the fine-grained grid search. Influenced by the analysis of guidance strength and fidelity/diversity in previous studies [2, 3, 4], we have employed this strategy to balance the trade-off between exploitation and exploration: Higher guidance strength focuses more on the modes of the guiding predictor, prioritizing fidelity over diversity, which is similar to exploitation. On the contrary, utilizing lower guidance strength reduces dependence on the guiding predictor, enabling increased engagement in exploration.
>
> - Please note that the guidance strength search processes and neural architecture sampling have minimal impact on the overall search time, as mentioned in the paper. The primary bottleneck of the search cost arises during the actual training of neural architecture candidates, where our proposed approach significantly reduces the number of architectures requiring actual training compared to existing methods.
>
> ---
> **References**
>
> [1] Exploring chemical space with score-based out-of-distribution generation, ICML, 2023.
>
> [2] Diffusion Models Beat GANs on Image Synthesis, NeurIPS 2021.
>
> [3] Classifier-Free Diffusion Guidance, NeurIPS Workshop 2021.
>
> [4] Diversity and Diffusion: Observations on Synthetic Image Distributions with Stable Diffusion, Arxiv, 2023.

---

> ### Author Response · Authors · 2023-11-16
> **Rebuttal by Authors (2/3)**
>
> **W1-2. Ablation studies on the impact of guidance strength.**
>
> - To empirically verify how this guidance strength affects the performance and further provide insights for future follow-up studies, we compared the neural architecture distributions generated using our sampling strategy (**balanced exploitation-exploration sampling strategy**), both when the guidance scale was **fixed at a large value** and when it was **fixed at a small value** (Please see **Table R2**). We conducted experiments on the CIFAR-10 dataset within the NB201 search space under the Transferable NAS scenario. The fixed large and small values correspond to the maximum and minimum guidance strengths in our strategy of gradually decreasing guidance strength with a scale 5000, respectively (i.e., 15000, 10000, 5000). As we mentioned in **Section 3.1** in the main paper, after generating neural architectures by each sampling strategy, we rank the generated architecture pools using a meta-learned dataset-aware predictor and also provide the average accuracy of the Top-5 neural architectures.
>
> **<Table R2>** Distribution of generated neural architectures across various sampling strategies. All methods generated neural architectures within the same sampling time.
> |                                            | Fixed at a large value (15000) | Balanced Exploitation-exploration sampling  (Ours) | Fixed at a small value (5000) |
> |:--------------------------------------------------:|:------------------------------:|:--------------------------------------------------:|:-----------------------------:|
> | Average Accuracy of Top-5 sorted architectures (%) |             93.262           |                     **94.156**                    |            93.978            |
> |               Standard Deviation (%)               |             0.395             |                       0.658                       |             0.953            |
>
>
> - As evident from **Table R2**, the average accuracy of Top-5 neural architectures generated with the **Fixed at a large value** strategy  (**93.262%**) is lower than the average accuracy of Top-5 neural architectures (**94.156%**) achieved by **Ours**, possibly due to heavily engaging in exploitation. On the other hand, fixing the guidance strength to one small value allows for greater engagement in exploration, leading to the generation of neural architectures with more diversity. However, since it guides towards regions far from the target sub-region, the average accuracy of the generated Top-5 neural architectures (**93.978%**) is lower than the result obtained by **Ours**, i.e., when the guidance strength is 0, it can be considered equivalent to random sampling.
> - Given the trade-off between exploration and exploitation depending on the guidance strength, we progressively adjust it during the generation of neural architectures. This sampling approach offers the advantage of balancing exploitation-exploration by gradually reducing the guidance strength.

---

> ### Author Response · Authors · 2023-11-16
> **Rebuttal by Authors (3/3)**
>
> **W2. Does the proposed approach have some model collapse phenomenon?**
>
> Thank you for providing valuable comments once again. As we explained in **W1-2**, using only one fixed large guidance strength to generate neural architectures may not lead to the generation of an optimal neural architecture. In other words, utilizing only one large guidance strength can lead to a heavy reliance on the guiding predictor (exploitation), which may result in phenomena corresponding to model collapse. However, by adopting our aforementioned sampling strategy that gradually decreases the guidance strength for generating neural architectures, we can achieve a balance between exploitation and exploration. As the guidance strength diminishes, the bias towards the predictor decreases, mitigating the risk of model collapse. Empirically, as seen in **Table R2**, the **Standard Deviation** values for the accuracy of the set of neural architectures generated by the fixed at a large value method and the proposed sampling method are **0.395** and **0.658**, respectively. The fixed at a large value method tends to produce relatively less diverse neural architecture structures. As a result, the proposed method (Top-5 accuracy: **94.154%**) outperforms the fixed at a large value method (Top-5 accuracy: **93.262%**).
>
> ---
> We appreciate the reviewer’s insightful and constructive comments. We incorporated the discussed points into **Section D.6 of the Appendix** in the revision to enhance the understanding of our guidance strength search strategy. These additions strengthen our paper, and we appreciate the reviewer's time and effort. If you have any further inquiries, please let us know.

---

> > ### Comment · Reviewer_qPhs · 2023-11-23
> >
> > Thank you very much for the detailed responses. My overall rating remains.

---

> > > ### Author Response · Authors · 2023-11-23
> > > **Thank you for your time and effort in reviewing our paper**
> > >
> > > Thank you for your acknowledgement.
> > >
> > > If you have any remaining concerns, please let us know. We are more than happy to address them.
> > >
> > > Best,
> > >
> > > DiffusionNAG Authors

---

> ### Author Response · Authors · 2023-11-20
> **Gentle reminder - Dear Reviewer qPhs**
>
> We appreciate your insightful and constructive comments once again. Would  you please review our response and revision? We have faithfully addressed your concerns and explained about the hyperparameter tuning rule for guidance strength. We would like to gently remind you about it. The following is a quick summary of the response.
> - We improved clarity by explaining the details of the hyperparameter tuning rule for guidance strength (Section D.6 of the Appendix).
> - We strengthened our method by conducting experiments on an ablation study for the hyperparameter tuning rule (Section D.6 of the Appendix).
>
> Please let us know if you have any other concerns so we can address them and if your concerns are addressed based on the rebuttal, we hope you update your rating accordingly. Once more, thank you for your helpful feedback.
>
> Best,
>
> DiffusionNAG authors

---

> ### Author Response · Authors · 2023-11-23
> **Gentle reminder - The interactive discussion phase will end in less than 10 hours**
>
> Dear Reviewer qPhs,
>
>
> We sincerely appreciate your time and effort in reviewing our paper. **As the interactive discussion phase will end in less than 10 hours, we politely ask you to check our responses.**  We have diligently responded to your comments, faithfully reflected them in the revision, and provided additional experimental results that you have requested.
>
> To help you quickly find what we did during the rebuttal period and further save more of your time, we briefly summarize our response below:
> ***
> * **Part 1 (1/3)**
>   * We explained our strategy for effectively defining the search range for guidance strength (Table R1 & Section D.6 of the Appendix).
> ***
> * **Part 2 (2/3)**
>   * We empirically verified how this guidance strength affects the performance (Table R2 & Section D.6 of the Appendix).
> ***
> * **Part 3 (3/3)**
>   * We provide an explanation of how we can mitigate the risk of the model collapse phenomenon alongside the experimental results.
> ***
> We hope that our responses have addressed your concerns, and we kindly hope you consider updating the rating accordingly. Please let us know if there are any other things that we need to clarify or provide. We sincerely appreciate your valuable suggestions.
>
> Best,
>
> DiffusionNAG Authors

---

### Official Review · Reviewer_qikk · 2023-11-09

**Soundness:** 3 good
**Presentation:** 3 good
**Contribution:** 3 good
**Rating:** 5
**Confidence:** 4

**Summary:**

The researchers are introducing a new approach called DiffusionNAG to overcome limitations in Neural Architecture Search (NAS) methods. Instead of traditional NAS, they propose a conditional Neural Architecture Generation (NAG) framework based on diffusion models. This framework, utilizing graph diffusion models, allows for efficient generation of task-optimal neural architectures with desired properties. DiffusionNAG incorporates parameterized predictors to guide the generation process, making it more efficient than previous NAS methods. The effectiveness of DiffusionNAG is demonstrated through experiments in Transferable NAS and Bayesian Optimization-based NAS scenarios.

**Strengths:**

1. This work treats neural architectures as directed graphs and introducing a graph diffusion model. It enables flexible generation of task-optimal architectures guided by parameterized predictors, surpassing the efficiency of traditional NAS methods.

2. DiffusionNAG outperforms baselines in Transferable NAS, achieving up to 20x speedups. Additionally, when integrated into a BO-based algorithm, it outperforms existing approaches, especially in the extensive MobileNetV3 search space on the ImageNet 1K dataset.

**Weaknesses:**

1. This work only reports results on two NAS benchmark search spaces. It is unclear whether it can achieve similar performance on much larger search spaces, such as the ones proposed in [1] and [2], which are widely used in previous works.

2. Fig. 3 and 4 compare results on existing AO strategies and various acquisition functions. However, there is no comparison to other methods on ImageNet, as shown in Tab. 1 and 2.

3. In Tab. 5, the authors include robust accuracy against the APGD attack, which is good. However, the corruption evaluation is insufficient, considering only glass blur. It would be better to include more comprehensive corruptions, such as those in ImageNet-C or CIFAR-10-C.

[1] Learning Transferable Architectures for Scalable Image Recognition, CVPR 2018.

[2] DARTS: Differentiable Architecture Search, ICLR 2019.

**Questions:**

Please address questions in "Weaknesses".

---

> ### Author Response · Authors · 2023-11-14
> **Rebuttal by Authors (1/3)**
>
> **W1. It is unclear whether DiffusionNAG can achieve good performance on much larger search spaces.**
> - Thank you for your valuable feedback. In response to the reviewer's concerns, we would like to clarify that **we have already conducted extensive experiments in the MobileNetV3 (MBv3) search space ($>10^{19}$), which is larger than DARTS ($\approx 10^{18}$) search space**. Through these experiments, we clearly demonstrated that DiffusionNAG achieved good performance in a larger search space (**Table 1** and **Figure 3** of the main paper).
>
> - **Background**: We provide a more detailed explanation of the MobileNetV3 (MBv3) search space. Recently, like the search spaces of NASNet or DARTS, the MBv3 search space has been widely utilized in the NAS community as one of the large search spaces [1-10]. Specifically, alongside NB201, the MBv3 search space serves as an established benchmark for our main experimental setup, Transferable NAS [4]. Therefore, we validated our approach in the same experimental setup with existing Transferable NAS methods to ensure a fair comparison. Additionally, this search space has been actively employed in Bayesian Optimization (BO)-based NAS [9].
> - **Experimental Results**: To save the reviewer’s time, we provide a summary of the experiments conducted for two key predictor-based NAS scenarios in the MBv3 search space as follows:
>   - **1) Transferable NAS**: DiffusionNAG achieved state-of-the-art (SOTA) performance on three out of four benchmark datasets (CIFAR-10, CIFAR-100, Aircraft, and Pets) within the MBv3 search space, as compared to the baselines (please refer to Table 1 in the main paper).
>   - **2) BO-based NAS**: We have shown that DiffusionNAG is an efficient and effective alternative to existing heuristic acquisition optimization strategies, such as random architecture sampling or architecture mutation, within the MBv3 search space (please refer to the left side and middle side of Figure 3 in the main paper). Specifically, the BO-based NAS approach employing DiffusionNAG as the acquisition optimization strategy surpassed existing BO-based NAS methods in experiments conducted on the ImageNet 1K dataset within the expansive MBv3 search space (please refer to the right side of Figure 3 in the main paper).
>
> - **Overview of the experiments we carried out**: Finally, we politely ask the reviewer to consider that we conducted extensive experiments as follows:
>   - 2 search spaces, including a large search space such as MBv3, which contains $\approx 10^{19}$ neural architectures.
>   - 5 different datasets, including a large-scale dataset, ImageNet 1K.
>   - 2 key predictor-based NAS scenarios on 2 search spaces.
>      - [Transferable NAS scenario] Transfer NAS experiments with a meta-learned dataset-aware predictor for 4 unseen datasets on both search spaces:  Main Paper Tables 1, 2.
>      - [BO-based NAS scenario] Improving existing BO-based NAS experiments for 2 datasets and 4 acquisition functions on both search spaces: Main Paper Figures 3, 4 & Appendix Figure 6.
>   - 4 ablation studies/analyses in the main paper.
>     - Analyzing the distribution of generated neural architectures: Main Paper Table 3 & Figure 2 & Appendix Figure 5.
>     - Analyzing the effectiveness of our proposed score network for neural architectures: Main Paper Table 4.
>     - Analyzing adaptation ability across different objectives: Main Table 5.
>     - Analyzing the distribution of operation types in generated neural architectures across different objectives: Appendix Figure 7.
>
>   We would be grateful if the reviewer could take these factors into account.
>
> ***
> #### **References**
>
> [1] Searching for mobilenetv3, ICCV 2019.
>
> [2] Once-for-all: Train one network and specialize it for efficient deployment, ICLR 2020.
>
> [3] NSGANetV2: Evolutionary multi-objective surrogate-assisted neural architecture search, ECCV 2020.
>
> [4] Rapid neural architecture search by learning to generate graphs from datasets, ICLR 2021.
>
> [5] HELP: Hardware-adaptive efficient latency prediction for nas via meta-learning, NeurIPS 2021.
>
> [6] Task-Adaptive Neural Network Search with Meta-Contrastive Learning, NeurIPS 2021.
>
> [7] Neural Architecture Transfer, TPAMI 2021.
>
> [8] CompOFA – Compound Once-For-All Networks for Faster Multi-Platform Deployment, ICLR 2021.
>
> [9] Transfer NAS with meta-learned bayesian surrogates, ICLR 2023.
>
> [10] MathNAS: If Blocks Have a Role in Mathematical Architecture Design, NeurIPS 2023.

---

> ### Author Response · Authors · 2023-11-14
> **Rebuttal by Authors (2/3)**
>
> **W2. In BO-based NAS scenarios (Figs 3 and 4), there is no comparison to other methods on ImageNet.**
>
> - We believe the reviewer may have overlooked **the right side of Figure 3**, which **contains comparison results with other baseline methods on ImageNet**. Specifically, we conducted a performance comparison between our proposed method and well-known NAS baseline methods such as BOHAMIANN [11], BONAS [12], EVOLUTION [13], LOCAL SEARCH [14], and RANDOM, highlighting our superiority.
> - Furthermore, as illustrated in **Table R1** below, to address the reviewer's concern, we enhance the comprehensiveness of our results by introducing new baselines in addition to those featured in the right subfigure of Figure 3, resulting in **7 baseline methods** evaluated on ImageNet 1K. DiffusionNAG consistently outperforms other methods, showcasing superior performance across the same **Trained Archs** (the number of architectures requiring full training for obtaining final accuracy). The final performance reaches **83.55%**, surpassing that of the existing methods.
>
>
> **<Table R1>**
> |    Method \ Trained Archs    |     40    |     60    |     80    |    100    |
> |:----------------------------:|:---------:|:---------:|:---------:|:---------:|
> | RANDOM                       |   82.51   |   82.58   |   82.60   |   82.72   |
> | GCN Pred.                    |   82.51   |   82.58   |   82.60   |   82.76   |
> | EVOLUTION                    |   82.68   |   83.00   |   83.15   |   83.21   |
> | LOCAL SEARCH                 |   83.21   |   83.30   |   83.36   |   83.43   |
> | BONAS                        |   82.71   |   82.98   |   83.09   |   83.25   |
> | DNGO                         |   82.87   |   83.09   |   83.25   |   83.49   |
> | BOHAMIANN                    |   82.92   |   83.21   |   83.36   |   83.44   |
> | **Ours (DiffusionNAG + BO)** | **83.29** | **83.43** | **83.52** | **83.55** |
>
>
> - We have updated **the right subfigure of Figure 3** by including additional baseline results in **the revision**.
>
> We appreciate the reviewer's comments since we believe that these additional results further strengthened our paper.
>
>
> ***
> #### **References**
> [11] Bayesian optimization with robust Bayesian neural networks, NeurIPS 2016.
>
> [12] Bridging the gap between sample-based and one-shot neural architecture search with bonas, NeurIPS 2020.
>
> [13] Regularized evolution for image classifier architecture search, AAAI 2019.
>
> [14] Exploring the loss landscape in neural architecture search, UAI, 2021.

---

> ### Author Response · Authors · 2023-11-14
> **Rebuttal by Authors (3/3)**
>
> **W3. It would be better to include more corruptions, such as those in ImageNet-C or CIFAR-10-C (Table 5).**
>
> - Thank you for your valuable comments, especially for the positive feedback on our experiments on generating robust neural architectures against APGD attacks. Additionally, following the reviewer’s suggestion, we conducted further experiments to explore a broader range of corruptions introduced not only in glass blur but also in various corruptions in CIFAR-10-C. As evident in **Table R2**, **we conducted experiments to find robust neural architectures for 14 diverse corruptions at severity level 2 in CIFAR-10-C**. Similar to **Table 5** of the main paper, we demonstrated the capabilities of DiffusionNAG to easily adapt to new tasks without retraining the score network. We achieved this by replacing task-specific predictors in a plug-and-play manner.
> - Specifically, we train a clean predictor, $f_{\text{clean}}$, to predict the clean accuracy of neural architectures using randomly sampled pairs of architectures and their corresponding CIFAR-10 clean accuracy from the Robustness-NB201 benchmark [15]. Afterward, we create the training dataset by including pairs of architectures and their respective robust accuracy for each corruption scenario from the Robustness-NB201 benchmark [15]. Then, we train the target corruption predictor, $f_{\text{corruption}}$ using the training dataset to predict the robust accuracy of neural architectures under specific corruption types.
> - With the guidance of the trained $f_{\text{clean}}$, we generate a pool of 20 architectures ($S_{\text{clean}}$).  Similarly, with the guidance of the task-specific predictors ($f_{\text{corruption}}$), we generate another pool of 20 architectures for each target corruption ($S_{\text{corruption}}$).
> Subsequently, we retrieve the robust accuracy on both $f_{\text{clean}}$-guided and  $f_{\text{corruption}}$-guided architecture pools ($S_{\text{clean}}$ and $S_{\text{corruption}}$) from the Robustness-NB201 benchmark [15] for each corruption scenario.
>
> We report the highest robust accuracy, as below:
>
> **<Table R2>**
>
> |                             |     brightness    |       contrast       |   defocus blur  | elastic transform |       fog      |   frost  | gaussian blur |             |
> |-----------------------------|:-----------------:|:--------------------:|:---------------:|:-----------------:|:--------------:|:--------:|:-------------:|:-----------:|
> | **$f_{\text{clean}}$**      |        89.3       |         58.9         |       64.6      |        61.0       |    **74.7**    | **60.2** |      30.5     |             |
> | **$f_{\text{corruption}}$** |      **89.4**     |       **59.6**       |     **64.9**    |      **61.4**     |      74.4      |   58.9   |    **49.6**   |             |
> |                             | **impulse noise** | **jpeg compression** | **motion blur** |    **pixelate**   | **shot noise** | **snow** | **zoom blur** | **average** |
> | **$f_{\text{clean}}$**      |        55.2       |         61.6         |       43.7      |        71.7       |      39.7      |   70.4   |      44.9     |     59.0    |
> | **$f_{\text{corruption}}$** |      **57.7**     |       **64.6**       |     **49.2**    |      **72.1**     |    **51.0**    | **71.2** |    **52.9**   |   **62.6**  |
>
> - As shown in **Table R2**, the maximum accuracies of the architecture pool generated using the predictor $f_{\text{corruption}}$ trained for the target corruption surpass those obtained using the accuracy predictor $f_{\text{clean}}$ trained for clean accuracy. For **12** out of the 14 corruptions, excluding fog and frost corruption, the maximum accuracy of the architecture pool guided by the $f_{\text{corruption}}$ predictor exceeds that of the pool guided by $f_{\text{clean}}$. The average maximum accuracy increased by **3.6%**, reaching **59.0%** and **62.6%**, respectively.
> - We included the experiments of **Table R2** in **Table 10 of Appendix Section D.5** in the revision.
>
> ---
> **Reference**
>
> [15] Neural architecture design and robustness: A dataset, ICLR 2023.
>
> ---
> We believe that our paper became much stronger after including the additional experiments you have suggested. Thank you for the constructive comments. Please inform us if there are any others you would like us to address. If you believe the concerns are addressed, we would appreciate it if you could adjust the rating accordingly.

---

> ### Author Response · Authors · 2023-11-20
> **Gentle reminder - Dear Reviewer qikk**
>
> The interactive discussion phase will end this Wednesday (22nd AOE), and we cannot have discussions with you anymore after the deadline. Would you please check our responses to your comments and the new experimental results you have requested? The following is a quick summary of the response.
>
> - We improved clarity by explaining that our paper included the experiments demonstrating the efficacy of the proposed method on the large search space (which is larger than DARTS) (Table 1 and Figure 3 of the main paper).
> - We improved clarity by explaining that our paper included the experiments comparing our method with other BO-based methods on ImageNet (Right side of Figure 3).
> - We strengthened our method by conducting comparisons with more BO-based methods (updated right side of Figure 3).
> - We strengthened our method by conducting experiments on additional corruptions (Appendix Section D.5).
>
> We would be grateful if you could offer feedback on this, and if your concerns are addressed based on the rebuttal, we hope you update your rating accordingly. We sincerely thank you for your valuable and insightful suggestion once again.
>
> Best,
>
> DiffusionNAG authors

---

> ### Author Response · Authors · 2023-11-23
> **Gentle reminder - The interactive discussion phase will end in less than 10 hours**
>
> Dear Reviewer qikk,
>
> We sincerely appreciate your time and effort in reviewing our paper. **As the interactive discussion phase will end in less than 10 hours, we politely ask you to check our responses.**  We have diligently responded to your comments, faithfully reflected them in the revision, and provided additional experimental results that you have requested.
>
> To help you quickly find what we did during the rebuttal period and further save more of your time, we briefly summarize our response below:
> ***
> * **Part 1 (1/3)**
>   * We clarified that DiffusionNAG achieved good performance in the large search space. (Table 1 & Figures 3, 4 of the main paper)
> ***
> * **Part 2 (2/3)**
>   * We conducted experiments showing that DiffusionNAG outperformed several BO-based NAS methods. (Table R1 & Figure 3 of the main paper)
> ***
> * **Part 3 (3/3)**
>   * We conducted experiments showing that DiffusionNAG generated robust neural architectures successfully under various corruption scenarios (Table R2 & Section D.5 of the Appendix).
>
> ***
> We hope that our responses have addressed your concerns, and we kindly hope you consider updating the rating accordingly. Please let us know if there are any other things that we need to clarify or provide. We sincerely appreciate your valuable suggestions once again.
>
> Best,
>
> DiffusionNAG Authors

---

### Author Response · Authors · 2023-11-19
**Global Response by Authors**

Dear Reviewers,

Firstly, we would like to express our gratitude to our reviewers for their efforts and time in reviewing our paper. The positive feedbacks that we received particularly encouraged us, summarized as:

- **Motivation and Novelty**: DiffusionNAG is **well-motivated** (****qPhs****), **novel** (****ZVno****, ****qikk****), and **interesting** (****EDN5****).
- **Diffusion Model for DAGs**: Especially, the proposed diffusion model for directed acyclic graphs is **needed** (**qPhs**), and **useful across various fields** (****ZVno****).
- **Predictor Guidance:** Moreover, the proposed guidance scheme is **well-suited** for NAS tasks (****qPhs****), **enables flexible generation** of task-optimal architectures (****qikk****), and **facilitates utilizing different predictors** dependent on tasks (****ZVno****).
- **Writing Quality**: The paper is **well-written** and **easy to follow** (****ZVno****).
- **Efficiency**: DiffusionNAG enables **faster search** (****ZVno****), **reducing NAS cost** (****EDN5****), and **surpassing the efficiency** of traditional NAS methods achieving **20$\times$ speedups** (****qikk****).
- **Effectiveness:** DiffusionNAG **outperforms** existing approaches (**qikk, EDN5, ZVno**).

Secondly, we are grateful for the creative suggestions provided by the reviewers, and we have diligently incorporated them into our revision (highlighted in blue). Most of the new content has been appended in the appendix to avoid interference with references during the rebuttal period, with plans to integrate it into the main text later.

Lastly, reviewers expressed concerns in the weakness section, primarily asking for clarification and discussions on potential future research. We have addressed each point individually, providing additional experiments where necessary to enhance the clarity of the paper. We summarize the details of these improvements below:

- **qikk:**
    - We improved clarity by explaining that our paper already included the experiments demonstrating the efficacy of the proposed method on the large search space (which is larger than DARTS) (Table 1 and Figure 3 of the main paper).
    - We improved clarity by explaining that our paper already included the experiments comparing our method with other BO-based methods on ImageNet (Right side of Figure 3).
    - We expanded experiments on additional BO-based methods (updated right side of Figure 3).
    - We expanded experiments on additional corruptions (Appendix Section D.5).
- **qPhs:**
    - We improved clarity by explaining the details of the hyperparameter tuning rule for guidance strength (Section D.6 of the Appendix).
    - We included experiments on an ablation study for the hyperparameter tuning rule (Section D.6 of the Appendix).
- **ZVno:**
    - We improved clarity by explaining details of prior works, specifically the meta-learned dataset-aware predictor (Section B.5 of the Appendix).
    - We improved clarity by explaining the details of our score network.
- **EDN5:**
    - We clarified our major challenges (difficult validity conditions of NAS tasks) and technical contributions of our score network.
    - We provided a discussion regarding the availability of using a discrete diffusion model instead of a continuous diffusion model for designing our DiffusionNAG.
    - We provided discussions for future works, including applying our DiffusionNAG to generate 1) optimal generative models and 2) sparse foundation models (Section E of the Appendix).

We believe that these updates address the concerns raised by the reviewers and contribute to the overall clarity and strength of our paper. We hope our response can help you in finalizing the scores of our paper. If you have any other questions, please feel free to reply, and we will answer them as soon as possible.

Sincerely,

DiffusionNAG Authors

---

### Author Response · Authors · 2023-11-23
**Gentle reminder - The interactive discussion phase will end in less than 4 hours**

**[Gentle reminder - The interactive discussion phase will end in less than 4 hours]**

Dear Reviewers,

Thank you all for your time and effort in reviewing our paper.

We would like to kindly remind you that **there are only 4 hours left for the interactive discussion.**

**Therefore, we politely ask you to review our responses and the revision.**

We hope our responses and the clarifications have addressed any remaining questions, and we are willing to address any further inquiries you may have.

Thank you for reviewing our paper once again.

Yours sincerely,

DiffusionNAG Authors.

---

### Meta-Review · Area_Chair_qDSF · 2023-12-07

**Metareview:**

The paper presents DiffusionNAG, a new method for finding neural network architectures using diffusion models. This approach is different because it sees these architectures as graphs and uses a special process for making them. It's more efficient than older methods, being faster and performing better. The researchers tested it in different settings, like Transferable NAS and Bayesian Optimization-based NAS, and found it works well for creating specific architectures for certain tasks.

**Justification For Why Not Higher Score:**

The reviewers expressed doubts about the originality of the paper, suggesting that it merely extends existing diffusion models to the Neural Architecture Search (NAS) problem. In response, the authors emphasized the challenge they faced in adapting these models to the NAS context. Specifically, they highlighted the difficulty of representing neural architectures as directed graphs, noting that existing graph diffusion models typically focus on undirected graphs. To address this, the authors developed a specialized score network to handle directed graph representations. However, this contribution is seen as a relatively straightforward technical improvement rather than a major innovation.

**Justification For Why Not Lower Score:**

The application of diffusion models to the Neural Architecture Search (NAS) problem, as explored in this paper, presents an intriguing concept. The authors have thoughtfully selected a conditional diffusion model as their foundational method and integrated a NAS performance predictor to tailor the conditions within these models. In implementing this approach, they have leveraged transferable NAS. While the core technical components of their methodology are extensively documented in existing literature, the authors' comprehensive analysis and careful selection of the most suitable modules to combine into a functional and potentially effective NAS method is noteworthy. This approach could offer valuable insights and inspiration to the broader NAS community.

---

### Decision · Program_Chairs · 2024-01-16

Accept (poster)